


# Extraction, purification, and clumped isotope analysis of methane ($\Delta^{13}CDH_3$ and $\Delta^{12}CD_2H_2$) from sources and the atmosphere

Malavika Sivan, Thomas Röckmann, Carina van der Veen, Maria Elena Popa

5   Institute for Marine and Atmospheric Research Utrecht (IMAU), Utrecht University, the Netherlands

*Correspondence to: Malavika Sivan (m.sivan@uu.nl)*

10 **Abstract**

Measurements of the clumped isotope anomalies ($\Delta^{13}CDH_3$ and $\Delta^{12}CD_2H_2$) of methane ($CH_4$) have shown potential for constraining $CH_4$ sources and sinks. Together with the bulk isotopic composition, they can be used to unravel the information about the formation and history of 15  $CH_4$. At Utrecht University, we use the Thermo Ultra high-resolution isotope ratio mass spectrometer to measure the clumped isotopic composition of $CH_4$ from samples of various origins such as geologic sites, biogenic systems, and laboratory incubation experiments, and from the atmosphere.

We have developed an extraction system with three sections for extracting and purifying $CH_4$ 20  from high (>1 %), medium (0.1-1 %), and low-concentration (< 1 %) samples, including atmospheric air (~2 ppm = 0.0002 %). Depending on the $CH_4$ concentration, a quantity of sample gas is processed that delivers $3 \pm 1$ mL of pure $CH_4$, which is the quantity typically needed for one clumped isotope measurement. For atmospheric air with a $CH_4$ mole fraction 25  of 2 ppm, we currently process up to 1100 L of air.

The analysis is performed on pure $CH_4$, using a dual inlet setup. The complete measurement time for all isotope signatures is about 20 hours for one sample. The mean internal precision of sample measurements is $0.3 \pm 0.1$ ‰ for $\Delta^{13}CDH_3$ and $2.4 \pm 0.8$ ‰ for $\Delta^{12}CD_2H_2$. The 30  long-term reproducibility, obtained from repeated measurements of a constant target gas, over almost 3 years, is around 0.15 ‰ for $\Delta^{13}CDH_3$ and 1.2 ‰ for $\Delta^{12}CD_2H_2$. The measured clumping anomalies are calibrated via the $\Delta^{13}CDH_3$ and $\Delta^{12}CD_2H_2$ values of the reference $CH_4$ used for the dual inlet measurements. These were determined through isotope equilibration experiments at temperatures between 50 and 450 °C.

Here, we describe in detail the optimized sampling, extraction, purification, and measurement technique followed in our laboratory to measure the clumping anomalies of $CH_4$ precisely and accurately. We also give an overview of the results of samples of various origins measured using this procedure.

40

## 1. Introduction





Atmospheric methane, CH₄, is the second most important anthropogenic greenhouse gas after
CO₂. The global warming potential of CH₄ is 28 times greater than that of CO₂ over a 100-
year period. Having a shorter lifetime of ~11 years Li et al. (2022) compared to CO₂ (Archer
et al., 2009), CH₄ responds faster to changes in its source and sink fluxes than CO₂. This also
means that CH₄ emission reduction measures can have a relatively faster effect on
atmospheric composition, reducing global warming. Global scale measurements of CH₄ mole
fractions show an increasing trend since pre-industrial times. The current global mean
atmospheric CH₄ mole fraction as of January 2023 is 1972 ppb while the estimated pre-
industrial values were 700-800 ppb (NOAA 2023). This long-term increase is mostly
attributed to anthropogenic emissions (IPCC 2022). Precise direct atmospheric measurements
have revealed significant shorter-term variations in the growth rate of atmospheric CH₄,
including stable levels in the early 2000s followed by an accelerating increase since 2007.
Various studies have attempted to attribute this temporal change to variations in the balance
between different CH₄ sources and atmospheric sinks (Nisbet et al., 2016; Schwietzke et al.,
2016; Kirschke et al., 2013; Turner et al., 2017; Saunois et al., 2020; Stevenson et al., 2022;
Rigby et al., 2017; Worden et al., 2017; Lan et al., 2021; Basu et al., 2022; Drinkwater et al.,
2023; Thanwerdas et al., 2022). However, these existing studies do not converge on the same
conclusion. This shows we don't fully understand the CH₄ cycle yet, which means that we
cannot predict its future behaviour confidently.

Major CH₄ sources are often separated into three categories according to the production
mechanism: biogenic (wetlands, cattle, lakes, landfills), thermogenic (natural gas, coalbed
CH₄, shale gas, etc) and pyrogenic (biomass burning, combustion of fossil fuels, etc.)
sources. The main CH₄ sink in the troposphere is photochemical oxidation by OH and Cl
radicals. Part of the CH₄ that reaches the stratosphere is removed by Cl and O($^1D$). About 10
% of the atmospheric CH₄ is taken up by surface sinks (Topp and Pattey, 1997).

A method commonly used to identify different sources and sinks of CH₄ is based on
measurements of its bulk isotopic composition, denoted as $\delta^{13}$C and δD. Each source has a
characteristic isotopic composition range as shown in Fig 1a, as a result of the isotopic
composition of the various substrates and the process-dependent isotopic fractionation during
CH₄ formation (Whiticar et al., 1986; Whiticar, 1999; Sherwood Lollar et al., 2006; Etiope
and Sherwood Lollar, 2013; Conrad, 2002; Kelly et al., 2022; Menoud et al., 2020). CH₄ from
all these sources contribute to atmospheric CH₄ with an expected isotopic composition of the
source mixture around -54 ‰ for $\delta^{13}$C and -290 ‰ for δD (Whiticar and Schaefer, 2007) (as
shown in Fig 1a). The sink reactions preferentially remove the lighter isotopologues of CH₄
from the atmosphere (Saueressig et al., 2001; Cantrell et al., 1990; Whitehill et al., 2017)
resulting in an enrichment of the heavier isotopes in the residual CH₄. The combined effect of
emissions from the various sources and removal by the different sinks lead to an overall
atmospheric CH₄ bulk isotopic composition of around -48 ‰ for $\delta^{13}$C and -90 ‰ for δD.
Many measurements have been performed to date, using analysis in the laboratory on
collected samples, and field-deployable instruments at various sites to study the variations in
atmospheric CH₄ (Menoud et al., 2020; Menoud et al., 2021; Menoud et al., 2022; Lu et al.,
2021; Beck et al., 2012; Fernandez et al., 2022; Röckmann et al., 2016b; Sherwood et al.,





2017). However, due to the overlap of some of the source signatures, it is not always possible to distinguish different sources of $CH_4$ using the bulk isotopes (Fig 1a).

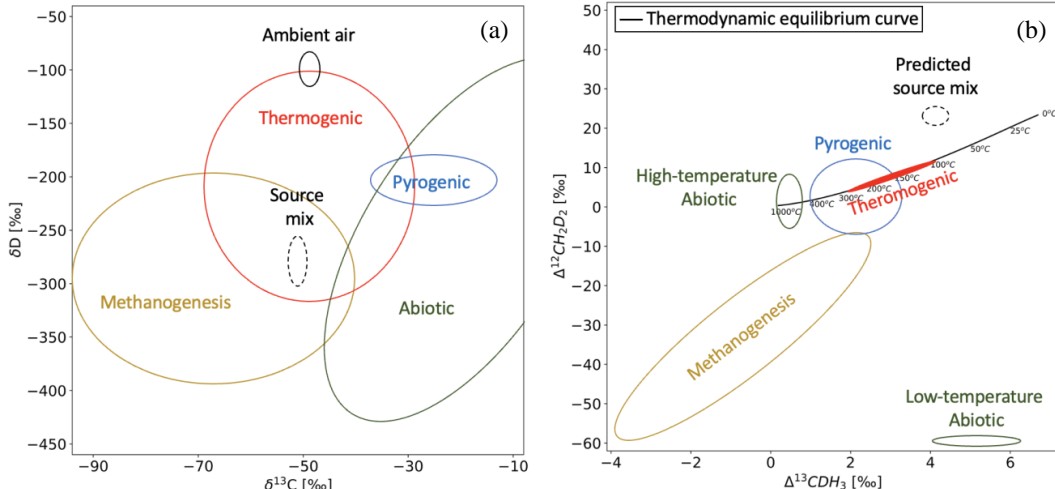


*Fig 1: An illustration of bulk (a) and clumped (b) isotopic composition of major $CH_4$ sources as reported so far.*

The measurement of the two most abundant clumped isotopologues ($^{13}CDH_3$ and $^{12}CD_2H_2$) of
$CH_4$ can be used as an additional tool to constrain $CH_4$ sources (Douglas et al., 2017; Eiler, 2007; Young et al., 2017; Stolper et al., 2014). The clumping anomalies, $\Delta^{13}CDH_3$ and $\Delta^{12}CD_2H_2$ are a measure of the degree of clumping together of heavier isotopes - $^{13}C$ and D & D and D, respectively, relative to the stochastic distribution of the light and heavy isotopes over all isotopologues of $CH_4$. At thermodynamic equilibrium, these anomalies are
temperature-dependent and can thus be used to calculate the $CH_4$ formation or equilibration temperature. In the case of thermodynamic disequilibrium, the clumped signatures can be exploited to identify various kinetic gas formation and fractionation (mixing, diffusion, etc.) processes. The clumped isotope signatures are specific to different sources and processes, independent of the bulk signatures, and thus can deliver additional information on sources
and cycling of $CH_4$ in the environment.

Measuring the clumped isotopic composition of $CH_4$, however, poses several technical challenges. The $^{13}CDH_3$ and $CD_2H_2$ molecules and $H_2O$ (which is always present in a mass spectrometer at much higher concentrations than the $CH_4$ clumped isotopologues) have very
slightly different masses, approximately 18.0409, 18.0439 and 18.0153 atomic mass units, respectively. This difference cannot be distinguished using a conventional mass spectrometer. Also, the $^{13}CH_4$ and $CDH_3$ have the same nominal mass (m/z 17), but these interferences can be circumvented by separating the C and H atoms, i.e., by converting the $CH_4$ to $CO_2$ for the $\delta^{13}C$ measurements, and to $H_2$ for $\delta D$. For clumped isotope measurements such an approach
would eliminate the signal we are looking for, thus the measurements need to be performed





on intact $CH_4$ molecules. In recent years, high-resolution isotope ratio mass spectrometers have become available that can resolve these small mass differences (Eiler et al., 2013; Young et al., 2017). These new instruments can separate the ion beams around m/z 18 corresponding to $CH_3D^+$, $^{12}CH_2D_2^+$ and $H_2^{16}O^+$ facilitating the $CH_4$ clumped isotope
measurements.

Another challenge includes the measurement of low ion currents and the instrument stability required for long measurement times. The ion currents of the $CH_4$ clumped isotopologues are very low, in our instrument typically around 6000 cps for $^{13}CH_3D^+$ and 100 cps for $^{12}CH_2D_2^+$. To achieve permil-level precision, the isotopologue ratios need to be measured for a long
time. This requires several mL of pure $CH_4$ for one measurement. To obtain pure-$CH_4$ for the measurements, the samples need to be purified. Isotope fractionation can occur during sample handling, extraction, and purification, potentially introducing biases and inaccuracies in the measured bulk and clumped isotopologue ratios. Careful consideration of sample preparation methods, including minimizing fractionation and optimizing purification procedures, is
crucial to ensure reliable and reproducible results. Another hurdle is that there are no readily available reference gases with known clumped isotopic composition to calibrate the measurements, so these need to be prepared.

A number of studies have reported the $\Delta^{13}CDH_3$ and $\Delta^{12}CD_2H_2$ of $CH_4$ from various sources,
e.g. natural gas seeps, rice paddies and wetlands, shale gas, coal mines, natural gas leakage, etc (Wang et al., 2015; Young et al., 2017; Stolper et al., 2018; Loyd et al., 2016). An overview of published clumped isotope signatures of $CH_4$ from different sources is illustrated in Fig 1b. Thermogenic $CH_4$ is usually formed in thermodynamic equilibrium and therefore lies on the thermodynamic equilibrium curve between 100-300 °C. Biogenic $CH_4$ production
is often characterised by dis-equilibrium $\Delta^{12}CD_2H_2$ values due to the kinetic isotopic fractionation associated with methanogenesis and/or combinatorial effects (Röckmann et al., 2016a). The reported range of values for abiotic and pyrogenic $CH_4$ is also shown in Fig 1b. A recent study has shown that we can expect enriched values for $CH_4$ remaining after the anaerobic oxidation of $CH_4$ (Giunta et al., 2022). Laboratory experiments have also been
performed to study clumping anomalies of $CH_4$ produced via different methanogenesis pathways (Ono et al., 2021) and abiotic environments (Young et al., 2017). The predicted clumping anomaly of the atmospheric $CH_4$ source mix resulting from the combination of all sources is about 4 ‰ for $\Delta^{13}CDH_3$ and 20 ‰ for $\Delta^{12}CD_2H_2$, as reported by Haghnegahdar et al. (2017).

Recent modelling studies have suggested the potential of clumped isotope measurements of atmospheric $CH_4$, especially $\Delta^{12}CD_2H_2$, to distinguish between the main drivers of change in the $CH_4$ burden (Chung and Arnold, 2021; Haghnegahdar et al., 2017). However, as mentioned above, the clumped isotope measurements require few mL of pure $CH_4$. Therefore, a challenge specific to atmospheric $CH_4$ measurements is the extraction of $CH_4$
from very large samples of air required (thousands of litres).



This paper describes the technical setups and procedures for $CH_4$ clumped measurements at Utrecht University including (i) the extraction and purification of $CH_4$ from high and low concentration samples, including the extraction from large quantities of air (~ 1000 L); (ii) calibration of measured anomalies using gas-equilibration experiments at different temperatures; (iii) the detailed settings and procedures of the actual isotope measurements using the Thermo Ultra mass spectrometer and (iv) the data processing and calculations involved. We also report the performance of these systems so far, in terms of precision, reproducibility, stability, etc. Thus, this paper serves as description of our measurement technique for future reference.

## 2. Methods

### 2.1 Notations, definitions, and calculations

The bulk isotopic composition of $CH_4$, denoted as $\delta^{13}C$ and $\delta D$, is defined as follows:

$$\delta^{13}C_{sample} = \frac{R^{13C}_{sample}}{R^{13C}_{VPDB}} - 1 \qquad \text{(Equation 1a)}$$

$$\delta D_{sample} = \frac{R^{D}_{sample}}{R^{D}_{VSMOW}} - 1 \qquad \text{(Equation 1b)}$$

where, $R^{13C}_{sample}$ and $R^{D}_{sample}$ are the isotopic ratios of $^{13}C/^{12}C$ and D/H of the sample and $R^{13C}_{VPDB}$ and $R^{D}_{VSMOW}$ are isotopic ratios of the international standards for $\delta^{13}C$ and $\delta D$ (VPDB and VSMOW) and their values are 0.011180 and 0.00015576 respectively (Assonov et al., 2020; Gonfiantini, 1978).

The clumped isotopic composition of $CH_4$ is expressed as clumping anomalies $\Delta^{13}CDH_3$ and $\Delta^{12}CD_2H_2$ relative to the clumped isotope ratio that would be obtained if the heavy isotopes $^{13}C$ and D were distributed randomly across all isotopologues in the same sample:

$$\Delta^{13}CDH_{3\,sample} = \frac{R^{13CD}_{sample}}{(4*R^{13C}_{sample}*R^{D}_{sample})} - 1 \qquad \text{(Equation 2a)}$$

$$\Delta^{12}CD_2H_{2\,sample} = \frac{R^{DD}_{sample}}{(6*(R^{D}_{sample})^2)} - 1 \qquad \text{(Equation 2b)}$$

$R^{13CD}_{sample}$ and $R^{DD}_{sample}$ are the isotopologue ratios of $^{13}CDH_3/^{12}CH_4$ and $^{12}CD_2H_2/^{12}CH_4$ of the sample and $R^{13C}_{sample}$ and $R^{D}_{sample}$ are isotope ratios of $^{13}C/^{12}C$ and D/H of the sample itself. The denominator in the Equations 2a and 2b give the expected random distribution of the clumped isotopologues in a sample, where 4 and 6 are symmetry factors. (Young et al., 2017)



## 2.2 Mass spectrometer specifications and measurement methods


CH$_4$ bulk and clumped isotopic compositions are determined using the Thermo Scientific Ultra HR-IRMS. The prototype of the instrument was introduced by Eiler et al. (2013) and the characteristics of the Thermo Ultra at Utrecht University have been explained in detail by Adnew et al. (2019). The instrument is operated with the advanced Qtegra™ software package, for data acquisition, instrument control, and data analysis.


The sample is introduced via one of the four variable volume bellows into the ion source and reference gas is provided from another bellow. After ionization in the ion source, the ion beam is accelerated, focused, and passed through a slit into the mass analyzer. Three different slit widths of 250,16, and 5 μm can be chosen in the standard setup, giving three resolution options: low (LR), medium (MR) and high resolution (HR), respectively. An additional 'aperture' option, which is an additional slit, can be turned on to trim the beam further, resulting in even higher resolution (HR+). However, increasing the resolution results in a decrease of intensity.



The ions are separated by energy and mass in the mass analyzer, which leads to very well focussed ion beams, and they are collected with a variable detector array that supports one fixed and eight moveable detector platforms, which are equipped with nine Faraday detectors (L1, L2, L3, L4, Center, H1, H2, H3, H4) that can be read out with selectable resistors with resistances between $3 \times 10^8\ \Omega$ and $10^{13}\ \Omega$. The three collector platforms at the high mass end (H2, H3 and H4) are additionally equipped with compact discrete dynode (CDD) ion counting detectors next to the Faraday detectors.


### 2.2.1   Characterization of the Ultra for CH$_4$ measurements


Clumped isotope measurements of CH$_4$ using the Ultra are performed at high resolution (5 μm entrance slit width) with aperture i.e., HR+ setting, to get the highest possible resolution. Two Faraday collectors are read out with resistors, $1 \times 10^{11}\ \Omega$ for $m/z$ 16 and $1 \times 10^{12}\ \Omega$ for $m/z$ 17-$^{13}$CH$_4$. To measure m/z 17-$^{12}$CDH$_3$ and the clumped isotopologues at m/z 18, we use the CDD of detector H4, which has a narrow detector slit. With careful tuning, the instrument can achieve mass resolving power higher than 42,000, which is sufficient to separate CH$_4$ isotopologues from each other, from contaminating isobars like H$_2$O$^+$, OH$^+$, NH$_3$$^+$, etc and the adducts formed in the source, $^{12}$CH$_5$$^+$, $^{13}$CH$_5$$^+$ and $^{12}$CDH$_4$$^+$.


As the high resolution is to a large degree achieved by using a very narrow source slit, most of the ions do not pass through the slit but deposit on the slit assembly. This leads to carbon accumulation around the slit and over time obstructs the passage of ions into the mass analyzer, resulting in reduced ion transmission and sensitivity. The carbon deposits can also introduce additional scattering and deflection of ions, leading to the broadening of mass peaks and decreased mass resolution. There can also be signal instabilities due to fluctuations in ion transmission. These effects together can compromise the instrument's capability to





resolve closely spaced ions. Therefore, we change the source slit regularly to avoid the impact of carbon deposits. To keep track of this, the number of counts of $^{12}CH_4^+$ of each measurement is monitored (Fig S1 in supplement). When the counts decrease to less than 0.5 times the counts of the first measurement using a new slit, the slit is replaced. The usual lifetime of one slit is around 6 months, depending on the number of $CH_4$ measurements done.

The main $CH_4$ isotopologues, $^{12}CH_4^+$, $^{13}CH_4^+$, $^{12}CH_3D^+$, $^{13}CH_3D^+$, and $^{12}CH_2D_2^+$ are measured in three different configurations on the Ultra. The configurations differ by the peak center mass setting and the relative distance between the detectors and the peak positions are finely adjusted (Fig 3) such that the right ions are detected by each detector. The details of the three different configurations, resistors and detectors used for the measurements on the Ultra are given in Table 1. In the first configuration, $^{12}CH_4^+$ (L1) and $^{12}CH_3D^+$ (H4-CDD) are measured for about 3 hours. The second configuration is set up to measure $^{12}CH_4^+$ (L3), $^{13}CH_4^+$ (L1), and $^{13}CH_3D^+$ (H4-CDD) and the third configuration to measure $^{12}CH_4^+$(L3), $^{13}CH_4^+$ (L1), and $^{12}CH_2D_2^+$ (H4-CDD). Configurations 2 and 3 are measured alternately for 18 hours in 7 cycles each lasting about 2.5 hours. Therefore, in total, one complete measurement of all three configurations takes about 20 hours. The sample and reference gases are measured alternately, each three times (= integrations) for a total of 201.3 seconds; the average of which is considered one data point. The result of one complete measurement is the average of all the data measured (outliers removed) and the internal precision is the standard error over these datapoints.

A summary of the natural abundances, molecular masses, expected intensity in cps (for AP613, the laboratory reference gas), and the counting statistics precision limit for all the five isotopologues are given in Table 2.

*Table 1: The details of the three different configurations, resistors and detectors used for the measurements on the Ultra.*

| Configuration | L3 width: 1.3 mm (amplifier) | L1 width: 0.6 mm (amplifier) | H4-CDD width: 0.04 mm | Center mass (Latest mass calibration) (amu) |
|---|---|---|---|---|
| 1:δD | | $^{12}CH_4^+$ $(10^{11}\ \Omega)$ | $^{12}CH_3D^+$ | 17.2612 |
| 2:δ$^{13}$C, Δ$^{13}$CDH$_3$ | $^{12}CH_4^+$ $(10^{11}\ \Omega)$ | $^{13}CH_4^+$ $(10^{12}\ \Omega)$ | $^{13}CH_3D^+$ | 18.4799 |
| 3: Δ$^{12}$CD$_2$H$_2$ | $^{12}CH_4^+$ $(10^{11}\ \Omega)$ | $^{13}CH_4^+$ $(10^{12}\ \Omega)$ | $^{12}CH_2D_2^+$ | 18.4825 |

*Table 2: A summary of the natural abundances, molecular masses, expected intensity in cps (for AP613, the laboratory reference gas), and the counting statistics precision limit for an integration time of 201.3 seconds for all the five isotopologues of CH₄ measured on the Ultra.*




| Isotopologue | Natural abundance (%) | Molecular mass | Intensity in cps (AP613) | Counting statistics (‰) |
|---|---|---|---|---|
| $^{12}CH_4$ | 98.88 | 16.0313 | $9e^8$ | $2.3e^{-03}$ |
| $^{13}CH_4$ | 1.07 | 17.034 | $9.5e^6$ | 0.023 |
| $^{12}CDH_3$ | 0.045 | 17.0376 | $5e^5$ | 0.099 |
| $^{13}CDH_3$ | $4.9e^{-04}$ | 18.0409 | 5000 | 0.99 |
| $^{12}CD_2H_2$ | $7.8e^{-06}$ | 18.0439 | 90 | 7.43 |

The gasses are measured at a source pressure of maximum 2.5 e$^{-7}$ mbar. The pressure in the source is controlled by the bellow pressure, which can be set and adjusted using Qtegra. The

typical pressure in the bellows required to achieve this source pressure for CH$_4$ is around 65-70 mbar. We use a continuous pressure adjustment method, which is, after each integration, the bellow pressures are checked 5 times, and the bellows are compressed by 0.1 mbar each time, until the set value is attained. The tolerance of the pressure adjustment is set to 0.5 mbar, so that the signal is stable within ± 0.7 %. This ensures that the instrument measures

the reference and sample at the same source pressure during the entire 20+ hours of measurement time.

All measurements are made relative to a reference gas, which is a stainless-steel canister filled from a high purity (>99.999%) CH$_4$ reference gas cylinder (AP613). The sample and

the reference are measured alternately, and then the bulk and clumped isotopic composition of the samples are calculated from the isotopologue ratios as follows:

$$\delta^{13C}_{sam-VPDB} = \delta^{13C}_{sam-ref} + \delta^{13C}_{ref-VPDB} + (\delta^{13C}_{sam-ref} * \delta^{13C}_{ref-VPDB}) \quad \textit{(Equation 3a)}$$


$$\delta^{D}_{sam-VSMOW} = \delta^{D}_{sam-ref} + \delta^{D}_{ref-VPDB} + (\delta^{D}_{sam-ref} * \delta^{D}_{ref-VSMOW}) \quad \textit{(Equation 3b)}$$

$$\Delta^{13CDH_3}_{sam} = \frac{(1+\delta^{13CDH_3}_{sam-ref})*(1+\Delta^{13CDH_3}_{ref})}{(1+\delta^{13C}_{sam-ref})*(1+\delta^{D}_{sam-ref})} - 1 \quad \textit{(Equation 3c)}$$


$$\Delta^{12CD_2H_2}_{sam} = \frac{(1+\delta^{12CD_2H_2}_{sam-ref})*(1+\Delta^{12CD_2H_2}_{ref})}{(1+\delta^{D}_{sam-ref})^2} - 1 \quad \textit{(Equation 3d)}$$

$\delta^{13C}_{sam-ref}$ , $\delta^{D}_{sam-ref}$, $\delta^{13CDH_3}_{sam-ref}$ and $\delta^{12CD_2H_2}_{sam-ref}$ are the values of the sample measured against the reference calculated from the measured ion intensities on the Ultra. These values are converted to the standard scales: $\delta^{13C}_{sam-VPDB}$, $\delta^{D}_{sam-VSMOW}$, $\Delta^{13CDH_3}_{sam}$ and $\Delta^{12CD_2H_2}_{sam}$ using the

formulae above. The clumping anomalies of the reference gas used for the measurements,



AP613, denoted as $\Delta_{ref}^{13CDH_3}$ $and$ $\Delta_{ref}^{12CD_2H_2}$, were assigned using temperature-equilibration experiments which are explained in detail in the next section. The bulk isotopic composition of AP613 denoted as $\delta_{ref-VPDB}^{13C}$ $and$ $\delta_{ref-VSMOW}^{D}$, was obtained by measurements using a conventional continuous flow IRMS system (Menoud et al., 2021).


### 2.3 Temperature calibration scale

To produce a CH₄-clumped isotope calibration scale, we performed a series of isotope exchange experiments at various temperatures. For this, the laboratory reference gas, AP613

was used, which is a commercially available pure CH₄ cylinder with known bulk isotopic composition. CH₄ from AP613 was equilibrated at temperatures ranging from 50 to 450 °C using two different catalysts: γ-Al₂O₃ for temperatures below 200 °C and Pt on Al₂O₃ for 200-450 °C.

Both catalysts were activated using the procedure explained in Eldridge et al. (2019). For each heating experiment, about 10 pellets of the catalyst were inserted in a 20 mL glass tube with a Teflon valve and evacuated to $10^{-3}$ mbar to remove adsorbed air and moisture. The tube was then filled with 140 mbar of pure O₂ and heated for about 5 hours at 550 °C for activation of the catalyst. After heating, the tube was evacuated overnight (12-14 hours) at

550 °C and then cooled to room temperature. The pellets were not exposed to outside air once activated. After the activated pellets were cooled to room temperature, 5-6 mL of pure CH₄ (AP613) was added to the tube and heated at the desired temperature and duration as given in Table 3.

The equilibrated gases were measured on the Ultra against the reference gas, i.e., unmodified CH₄ from the AP613 cylinder. The raw $\Delta^{13}CDH_3$ and $\Delta^{12}CD_2H_2$ values are calculated using equations 3c and 3d but assuming $\Delta_{ref}^{13CDH_3}$ and $\Delta_{ref}^{12CD_2H_2}$ to be zero. The raw values obtained in this way showed the expected dependence on temperature but with a shift due to the real clumped values of the reference being different from zero. To determine this offset, the

functions from Eldridge et al. (2019) were fit to the data with an added free parameter for the offset as given in equations 4a and 4b:

$$\Delta^{13}CDH_3 = a + \frac{1.47348\,x\,10^{19}}{T^7} - \frac{2.08648\,x\,10^{17}}{T^6} + \frac{1.1981\,x\,10^{17}}{T^5} - \frac{3.54757\,x\,10^{12}}{T^4} + \frac{5.54476\,x\,10^9}{T^3}$$
$$- \frac{3.49294\,x\,10^6}{T^2} + \frac{8.8937\,x\,10^2}{T} \qquad \qquad (Equation\ 4a)$$


$$\Delta^{12}CD_2H_2 = b - \frac{9.67634\,x\,10^{15}}{T^6} + \frac{1.71917\,x\,10^{14}}{T^5} - \frac{1.24819\,x\,10^{12}}{T^4} + \frac{4.30283\,x\,10^9}{T^3} - \frac{4.4866\,x\,10^6}{T^2}$$
$$+ \frac{1.86258\,x\,10^3}{T} \qquad \qquad (Equation\ 4b)$$



The parameters *a* and *b* were then optimized, keeping the shape of the temperature dependence constant, and were used to estimate the $\Delta^{13}CDH_3$ and $\Delta^{12}CD_2H_2$ values of our reference gas. In practice, this was done using a Monte Carlo simulation with 1000 runs: at each run, each data point was independently applied a random error based on the uncertainty of that measurement, assuming Gaussian distribution of the errors. The functions above were

then fitted, and a set of free parameters (*a* and *b*) were obtained. The final absolute $\Delta^{13}CDH_3$ and $\Delta^{12}CD_2H_2$ values of the reference were calculated by averaging the *a* and *b* parameters for all runs (with outliers removed) and the errors reported are the corresponding standard deviations.

**2.4 CH₄ extraction and purification system**

The schematic of the extraction system is shown in Fig 2:

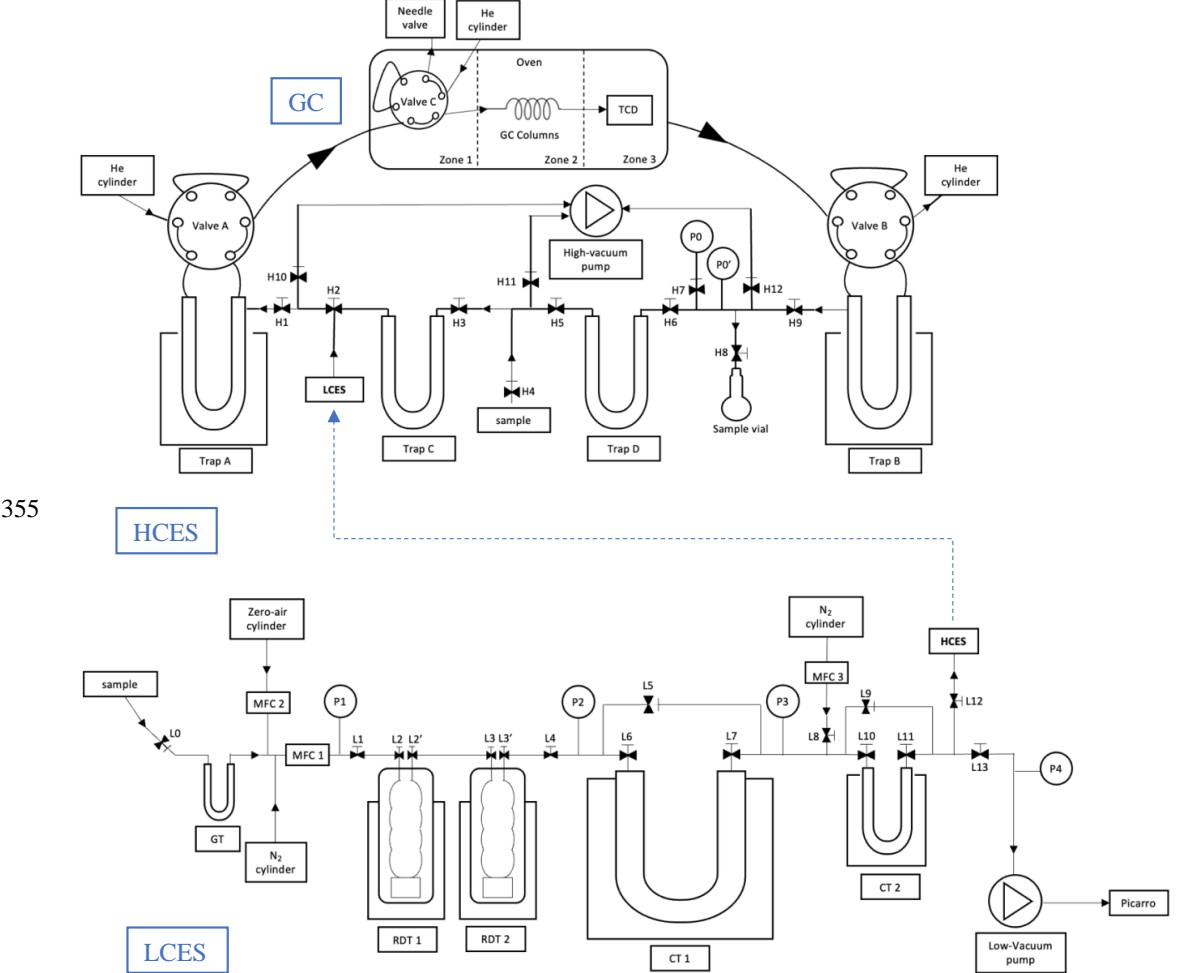





*Fig 2: Schematic of high-concentration (HCES) and low-concentration (LCES) extraction*
*system and the GC setup at IMAU. Samples are introduced to the HCES via H4 and to the*
*LCES via L0. The pre-concentrated sample in CT2 is transferred to Trap A via a connection*
*between L12 and H2.*


The bulk and clumped isotopic composition of $CH_4$ measured on the Ultra requires about $3 \pm 1$ mL of pure $CH_4$ for a single measurement. The $CH_4$ extraction and preconcentration procedure followed in our laboratory involves several steps depending on the sample concentration as explained below.


**2.4.1 HCES**

The high-concentration extraction system (HCES) is used to extract $CH_4$ from samples with more than 1 % of $CH_4$ i.e., extracting from up to 200 mL of sample gas. The HCES includes

two empty traps (Trap C and Trap D), two traps filled with Silica gel (Trap A and Trap B), and a gas chromatograph (GC) with a passive Thermal Conductivity Detector (TCD), all connected with ¼'' SS tubing and 316L VIM-VAR Swagelok valves. All the parts are shown in the schematic above (Fig 2). This system is built following the one described in Young et al. (2017).


The $CH_4$ in the sample gas is separated from the other components by GC, and then collected cryogenically on silica gel. The sample is introduced via valve H4 and collected in Trap A with Si-gel cooled to -196 °C with LN2. The pressure in the system is monitored to ensure that all the sample is trapped. The sample in Trap A is introduced into the GC from Trap A

using He at a flow rate of 30 mL/min for 5 min by warming the trap to about 70 °C using a hot water bath.

The GC has two columns used in series for the final purification of $CH_4$. A 5-meter ¼'' OD SS column packed with 5A° molecular sieve to separate $H_2$, Ar, $O_2$ and $N_2$ from

hydrocarbons and a 2-meter ¼'' OD SS column packed with HayeSep D porous polymer to separate $CH_4$ from the remaining higher hydrocarbons like $C_2H_6$, $C_3H_8$, etc. Wide columns of ¼'' are used to attain separation of more than 5 mL of $CH_4$ within 55 min.

$CH_4$ elutes from the GC column after $O_2$, $N_2$, and Kr. For concentrated samples (>5 % $CH_4$ in

air) without Kr, $O_2$ elutes around 10 min, $N_2$ around 22 min and $CH_4$ around 40 min when the GC is operated at 50 °C. After the elution of $N_2$ (35 min), Trap B with Silica gel is cooled with $LN_2$ to collect $CH_4$ for about 15 min. Once all the $CH_4$ is collected, Trap B is evacuated for 10 min to remove the He carrier gas while the trap is still cooled with $LN_2$. Following this, $CH_4$ is released from the Trap B by warming the trap to ~ 70 °C (hot water bath) and

collected in a sample vial filled with Silica gel and cooled with $LN_2$ to be transferred to the mass spectrometer.



For samples with concentrations between 1-5 % $CH_4$ in air, the $O_2$ and $N_2$ peaks are not resolved due to the larger sample size of the bulk air and form one overlapping peak on the chromatogram. This large peak is not fully separated from $CH_4$ either. $CH_4$ along with traces of $O_2$ and $N_2$ eluted from the GC is collected in Trap A instead of the sample vial and passed through the GC a second time for further purification (same steps as above). In the second round of extraction, the $O_2$ and $N_2$ peaks are small and well separated from each other and from the $CH_4$ peak. For samples with Kr (notably atmospheric samples), separation of pure $CH_4$ from Kr was only achieved when the GC columns were heated at 40 °C instead of 50 °C normally used for other samples. The comparison of chromatograms before and after Kr separation was achieved is shown in Fig 9.

After each chromatographic separation, the GC columns are baked at 200 °C for 30 min with He flow to remove $CO_2$, the heavier hydrocarbons, and other impurities. After baking, the columns are slowly cooled to 50 °C for the next extraction. Traps A and B are heated overnight at 150 °C while pumping with a high vacuum pump. The silica gel flask used for sample collection is evacuated until the next use.

### 2.4.2 LCES

Extracting $CH_4$ from large quantities of air involves stepwise increase of the $CH_4$ concentration by cryogenically trapping the sample gas in successively smaller charcoal traps, until the concentration is high enough for the sample to be further processed with the HCES. The low-concentration extraction system (LCES) is made of a 1/2" glass tube with J. Young high-vacuum PTFE valves and the major components are an empty glass trap (GT), two Russian Doll Traps (RDT1 and RDT2), and two charcoal traps (CT1 and CT2) as shown in Fig 2. A part of LCES is from the extraction system that has been used previously for CO isotope analysis (Bergamaschi et al., 2000; Bergamaschi et al., 1998).

The GT and RDTs are respectively used to remove $H_2O$ and $CO_2$ from air. This is followed by two pre-concentration steps in CT1 and CT2, which both collects all the $CH_4$ but only a small part of bulk air so that the $CH_4$ concentration increases in each step. The exhaust of the low-vacuum pump which draws the air though the extraction system is connected to a G2301 greenhouse gas analyzer (Picarro Inc.) to monitor $CO_2$, $CH_4$, and $H_2O$ concentrations during the whole extraction procedure. This ensures that a potential breakthrough is detected.

The air taken directly from outside or from a cylinder is first dried using GT cooled to -70 °C with a dry ice - ethanol slurry. A $Mg(ClO_4)_2$ tube after GT further dries the air sample before it is introduced to the traps for collection. RDT1 and RDT2, both cooled to -196 °C with $LN_2$ and connected in series, are used to scrub $CO_2$, $N_2O$, $H_2O$ traces and other condensable gases from the air. The $CO_2$-free air is then passed through CT1 (-196 °C) which traps $CH_4$ quantitatively, and only part of the remaining air components ($O_2$, $N_2$, etc). During this CT1 collection period, CT2 is bypassed. The flow of air is controlled using a Mass Flow Controller (MFC 1) and is adjusted to 6-6.5 L/min to maintain a pressure lower than 230 mbar in the glass line between L1 and L6 to avoid condensation of $O_2$ in the traps cooled with



LN$_2$, which is a potential danger. The glass line is partially heated using heating wires to avoid freezing of tubes and valves.

Once a quantity of about 1100 L of air is processed, the remaining air in the glass line is pumped until PS4 drops to 4 mbar. To transfer the collected air from CT1 to CT2, the LN$_2$ around CT1 is replaced with dry ice + EtOH slurry to warm the trap to -70 °C. At this temperature, the emerging N$_2$ + O$_2$ mixture is pumped out for 3-4 min, while the CH$_4$ stays in the CT1 trap. In the meantime, the bypassed CT2 is cooled to -196 °C with LN$_2$. The

remaining gas mix in CT1 is released by removing the dry ice slurry and heating CT1 with a hot water bath and is passed through CT2 (-196°C). As the pressure in the line drops to 10 mbar, 0.5 L/min of additional pure N$_2$ is used to transfer any remaining gas from CT1 to CT2 for 5 min via MFC 1. After this, the LN$_2$ bath of CT2 is replaced with dry ice + EtOH slurry and pumped for 1-2 min to further concentrate the air mixture. At the end of this step, the

final sample volume is less than 100 mL, and the sample can be transferred to Trap A of the HCES cooled with LN$_2$. CT2 is heated using a water bath and, after the pressure reading on PS3 drops to 0 mbar, it is flushed with pure N$_2$ from MFC 3 (at 5 mL/min for 2 min) to transfer the remaining gas. Once all the sample is collected in Trap A, the high-concentration extraction procedure is followed as explained above.


For samples with medium concentrations (0.1-1 % CH$_4$) i.e., < 3 L total sample volume, the first few steps of LCES are skipped and the sample is directly trapped in CT2. The remaining procedure is the same as explained above.

Before each extraction, RDTs and CTs are cleaned using 0.5 L/min of pure N$_2$ for 40 min while heating them with hot water baths at 70 °C to avoid contamination from the previous sample.

### 2.4.4 Extraction system tests with laboratory reference gas


The extraction and purification system was tested using three of our laboratory reference gases: AP613, CAL1549 and IMAU-3. Various mixtures of pure-AP613 in zero air and pure-CAL1549 in zero air were used to test the extraction system, and then the extracted CH$_4$ was measured on the Ultra. The separation of Kr from CH$_4$ in the GC and the effect on Kr on the

isotope measurements on the Ultra were tested using a 1:1 mixture of IMAU-3 and pure Kr.

To replicate the atmospheric CH$_4$ samples, pure-AP613 was mixed with zero air to a mole fraction of 2.5 ppm of methane in 1000 L. Since zero air is devoid of CO$_2$ and H$_2$O, GT and RDT2 were bypassed for these tests. RDT1 was still immersed in LN$_2$ to ensure that even

small traces of CO$_2$ were trapped and to check that the RDTs do not influence the clumping anomalies of CH$_4$. The rest of the procedure was followed as for normal sampling.

### 2.5 Quality checks for the Thermo Ultra





To establish the accuracy of the Ultra measurements, the Ultra δD and δ¹³C measurements
are compared to conventional bulk isotope measurements. Most samples are analysed for δD
and δ¹³C before the extraction and purification, using an independent conventional bulk
isotope measurement system (Menoud et al., 2020), and the results are compared to the ones
obtained from the Ultra measurements after the extraction.


Weekly "zero enrichment" measurements (same gas in both bellows) are done to check for
systematic difference between the bellows (e.g., by contamination, leaks, etc). These,
together with regular measurements of the pure CAL1549 gas, are used to monitor the
stability of the instrument and the reproducibility of the measurements. The internal precision

of the measurements is estimated for each measurement (sample or test gas) from the 1 σ
standard error over the whole measurement.

An inter-laboratory comparison with the the Nu Panorama high-resolution mass spectrometer
operated at University of Maryland (UMD) was done for the three laboratory reference gases:

AP613, CAL1549 and IMAU-3. The results of these comparisons are presented in the next
section.

## 3.  Results and Discussion

**3.1 Thermo Ultra measurements**

As described in section 2.2, clumped isotope measurements on the Ultra involve measuring
the different isotopologues in three configurations for a total of 20 hours. Typical mass scans
of the three configurations are shown in Fig 3. The position of the peak centers (marked with

red doted lines in Fig 3) is quite stable during the entire measurement procedure and small
mass shifts are corrected every hour using the peak center correction feature in the software.

**Configuration 1: $^{12}CH_4$ and $^{12}CDH_3$**

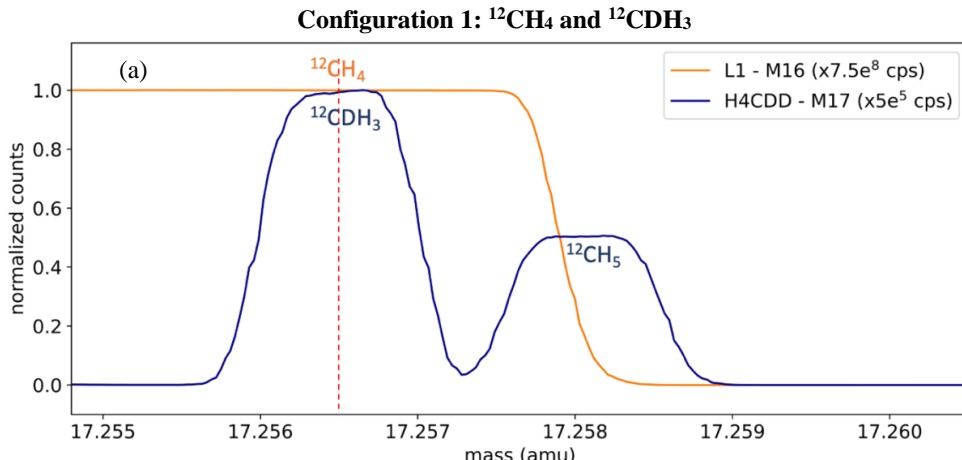




**Configuration 2: $^{12}CH_4$, $^{13}CH_4$ and $^{13}CDH_3$**

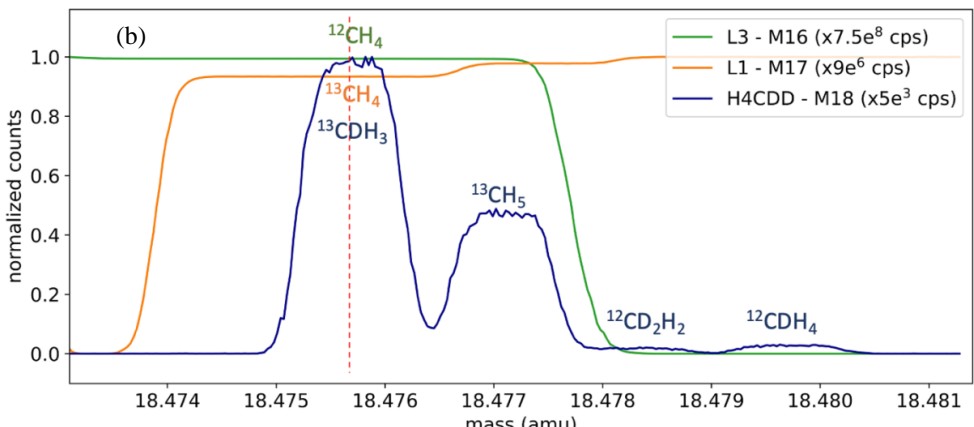

**Configuration 3: $^{12}CH_4$, $^{13}CH_4$ and $^{12}CD_2H_2$**

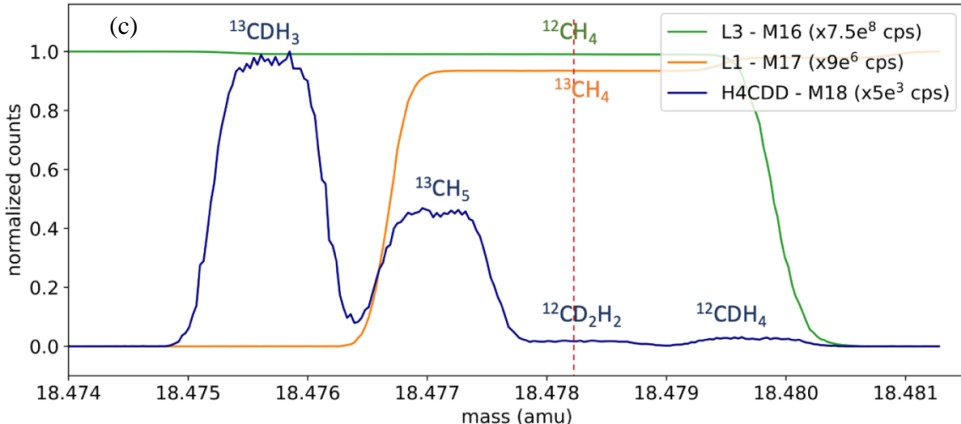


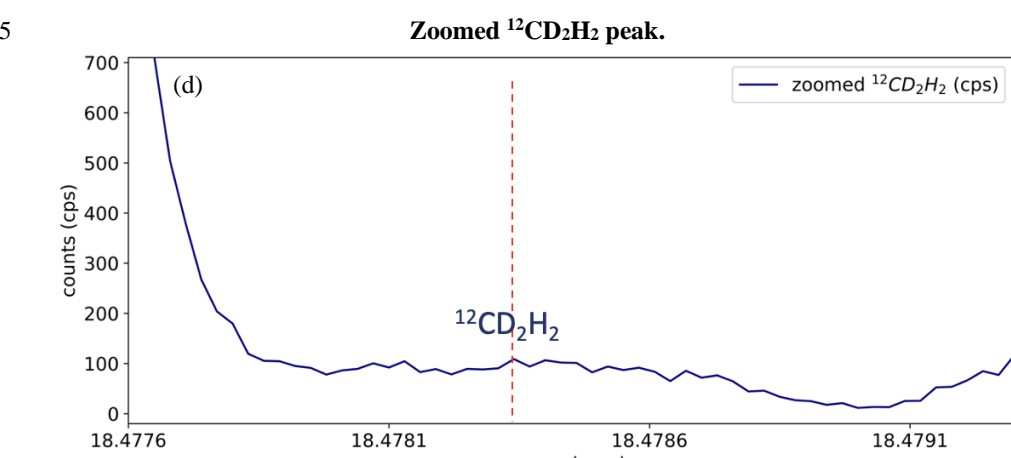

*Fig 3: Mass scans of three configurations to measure $^{12}CDH_3$ (a), $^{13}CH_4$ and $^{13}CDH_3$ (b),
$^{13}CH_4$ and $^{12}CD_2H_2$ (c and d). The x-axis values correspond to the peak center setting i.e.,
mass-17 in (a) and mass-18 in (b-d) and the other detectors are offset to these values to show*

*the other isotopologues on the same scale. The different detectors used, and the
normalization factors are given in the legends. The red dashed line indicates the peak center
mass setting. (d) shows the zoomed peak of $^{12}CD_2H_2$ and the counts measured.*

### 3.2 Temperature equilibration experiments


The results of the heating experiments are presented in Table 3. The equilibrated gas was
measured against non-equilibrated gas from AP613, which is the Ultra reference gas. Raw
measurement values relative to the reference gas are reported as $\Delta^{13}CDH_3$ raw and $\Delta^{12}CD_2H_2$
raw.


*Table 3: Summary of the equilibrated gas experiments, $\Delta^{13}CDH_3$ raw and $\Delta^{12}CD_2H_2$ raw
values are relative to the reference gas and $\Delta^{13}CDH_3$ absolute and $\Delta^{12}CD_2H_2$ absolute are
calculated using the assigned anomalies of the reference gas.*



| Temp (°C) | Catalyst | Duration (h) | $\Delta^{13}CDH_3$ (raw) (‰) | $\Delta^{13}CDH_3$ (absolute) (‰) | se | $\Delta^{12}CD_2H_2$ (raw) (‰) | $\Delta^{12}CD_2H_2$ (absolute) (‰) | se |
|---|---|---|---|---|---|---|---|---|
| 50 | $\gamma$-Al$_2$O$_3$ | 624 | 3.17 | 5.44 | 0.4 | 10.73 | 13.49 | 1.7 |
| 150 | $\gamma$-Al$_2$O$_3$ | 66 | 0.86 | 3.13 | 0.3 | 4.81 | 7.56 | 2.3 |
| 250 | Pt/Al$_2$O$_3$ | 120 | -0.31 | 1.95 | 0.3 | 4.02 | 6.77 | 2.6 |
| 300 | Pt/Al$_2$O$_3$ | 64 | -0.69 | 1.57 | 0.3 | 0.97 | 3.71 | 2.0 |
| 350 | Pt/Al$_2$O$_3$ | 144 | -0.64 | 1.62 | 0.3 | -2.44 | 0.29 | 2.4 |
| 400 | Pt/Al$_2$O$_3$ | 108 | -1.14 | 1.12 | 0.2 | -0.08 | 2.66 | 1.6 |


The measured values of heated AP613 at different temperatures were compared to the theoretical equilibrium curve, and the $\Delta^{13}CDH_3$ and $\Delta^{12}CD_2H_2$ values of AP613 were estimated using the Monte Carlo simulations as described in Sect. 2.3. The $\Delta^{13}CDH_3$ and

$\Delta^{12}CD_2H_2$ assigned to our reference gas, AP613 are: $\Delta^{13}CDH_3 = 2.23 \pm 0.12$ ‰ and $\Delta^{12}CD_2H_2 = 3.1 \pm 0.9$ ‰. Since this pair of values for the clumping anomalies doesn't lie on the thermodynamic equilibrium curve, we cannot assign a formation temperature value to AP613. The absolute values of $\Delta^{13}CDH_3$ and $\Delta^{12}CD_2H_2$ calculated using the assigned values of AP613 are given in Table 3 and in Fig 4.


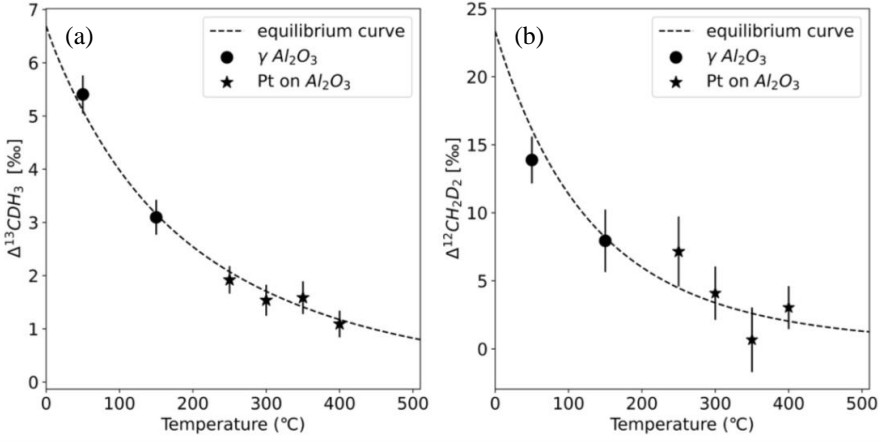

*Fig 4: Absolute $\Delta^{13}CDH_3$ and $\Delta^{12}CD_2H_2$ of the equilibrated gas compared to the theoretical equilibrium curve, calculated using the assigned anomalies of the reference gas, AP613:*
*$\Delta^{13}CDH_3 = 2.23 \pm 0.12$ ‰ and $\Delta^{12}CD_2H_2 = 3.1 \pm 0.9$ ‰. The data points represent the equilibrated gas at different temperatures with the markers corresponding to the different catalysts as given in the legend. The black dashed line is the thermodynamic equilibrium curve.*



### 3.3 Internal precision and reproducibility of the Ultra measurements


The average standard error of the measured $\delta^{13}C$, $\delta D$, $\delta^{13}CDH_3$ and $\delta^{12}CD_2H_2$ values and its comparison to the expected precision based on counting statistics of the shot noise are given in Table 4. Achieved precisions are very close to the shot noise limit for $\delta^{13}C$, $\delta^{13}CDH_3$ and

$\delta^{12}CD_2H_2$. Typically, $\delta D$ measurements are about 2 times worse than the shot noise limit. This may be because of the high-count rates (order of $10^5$ cps) measured using the H4-CDD detector, which has a narrow collector slit. However, the changes in $\delta D$ between different samples are much higher than the achieved precision, which is better than the one for conventional CF-IRMS instruments.


The average precision (1 $\sigma$ standard error) of calculated clumping anomalies of over 300 measurements in the last 3 years, is $0.3 \pm 0.1$ ‰ for $\Delta^{13}CDH_3$ and $2.4 \pm 0.8$ ‰ for $\Delta^{12}CD_2H_2$ depending on the $CH_4$ sample volume and measurement duration. The precision of $\Delta^{13}CDH_3$ and $\Delta^{12}CD_2H_2$ is calculated by propagating the error from the measured $\delta^{13}C$, $\delta D$, $\delta^{13}CDH_3$

and $\delta^{12}CD_2H_2$ values, using the equations 3c and 3d.

*Table 4: Average standard errors of $\delta^{13}C$, $\delta D$, $\delta^{13}CDH_3$ and $\delta^{12}CD_2H_2$ measurements on the Ultra and the expected errors from counting statistics of the shot noise. The "factor worse" shows how good our measurements are compared to the shot noise limit.*


| $\delta$ measured on the Ultra | Expected error (‰) | Actual error (‰) | Std dev of error (‰) | Factor worse |
|---|---|---|---|---|
| $\delta^{13}C$ | 0.006 | 0.007 | 0.002 | 1.16 |
| $\delta D$ | 0.045 | 0.110 | 0.03 | 2.4 |
| $\delta^{13}CDH_3$ | 0.293 | 0.312 | 0.05 | 1.06 |
| $\delta^{12}CD_2H_2$ | 2.22 | 2.26 | 0.8 | 1.03 |

The measurement procedure is slightly modified for samples smaller than 2 mL of $CH_4$. In such cases, $^{12}CD_2H_2$ is measured relatively longer than the standard procedure, with shorter measurements of $^{12}CDH_3$ to attain the maximum possible precision for $\Delta^{12}CD_2H_2$.






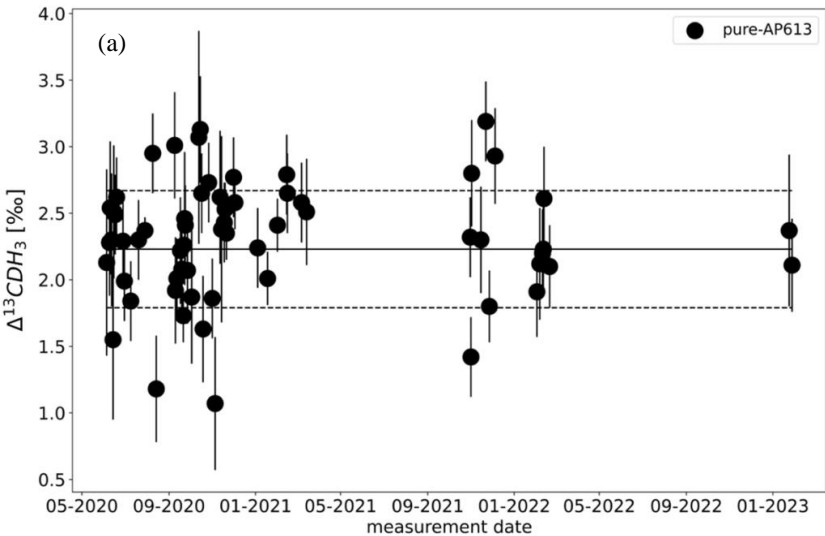

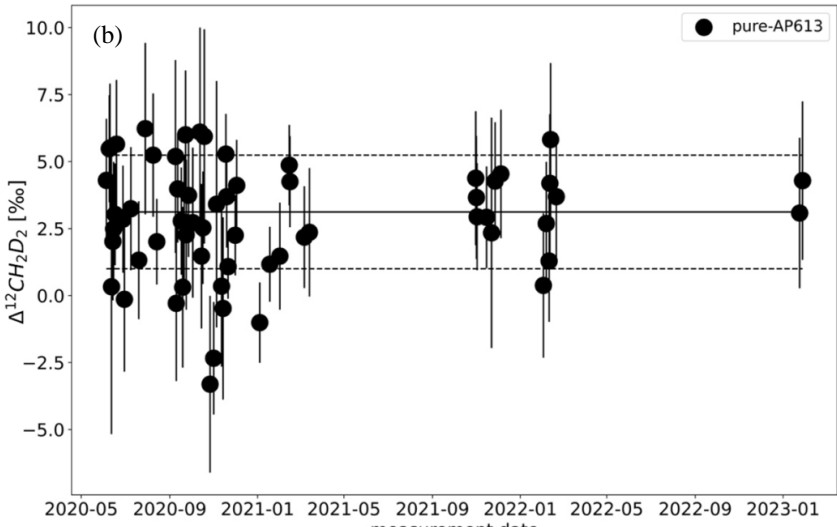

*Fig 5: Results of the zero enrichment measurements, each dot representing the calculated clumping anomalies $\Delta^{13}CDH_3$ (a) and $\Delta^{12}CD_2H_2$ (b) of gas AP613. The solid black line represents the values of AP613 assigned from the temperature calibration experiments and the black dashed lines indicate the $1\sigma$ std.dev of these measurements over 3 years.*

The results of the zero enrichment measurements using AP613 are shown in Fig 5. The values of AP613 calculated as samples from the zero enrichment measurements fall symmetrically around the known value, i.e., there is no systematic difference. The mean of these measurements done over 3 years is $2.3 \pm 0.1$ ‰ for $\Delta^{13}CDH_3$ and $3.2 \pm 0.3$ ‰ for $\Delta^{12}CD_2H_2$ and this is comparable to $2.2 \pm 0.1$ ‰ and $3.1 \pm 0.9$ ‰ for $\Delta^{13}CDH_3$ and $\Delta^{12}CD_2H_2$



respectively, obtained from the heating experiments (details in sect 3.2). The standard
       deviation of these measurements, 0.4 ‰ for $\Delta^{13}CDH_3$ and 2.1 ‰ for $\Delta^{12}CD_2H_2$, is close to the
       typical measurement error, which shows that there are no other large sources of errors in the
       sample measurements (e.g., leaks in the inlet, room temperature variations etc) and that both
       bellows used for the measurements behave similarly.


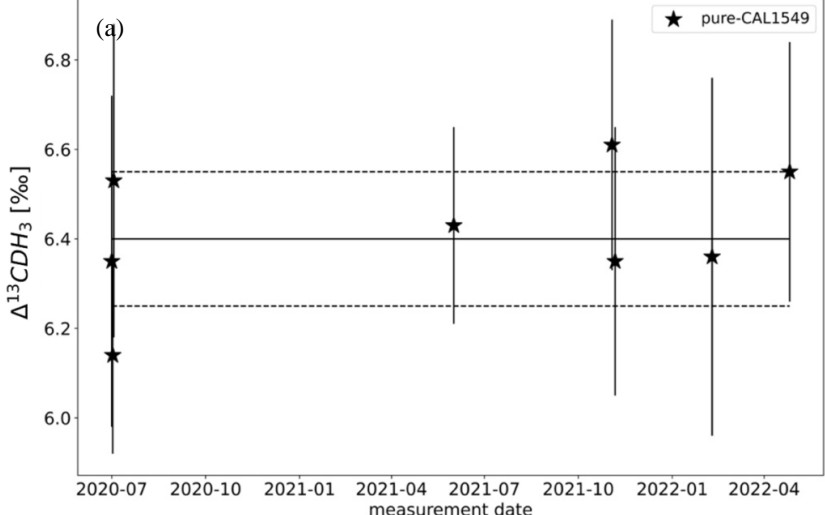

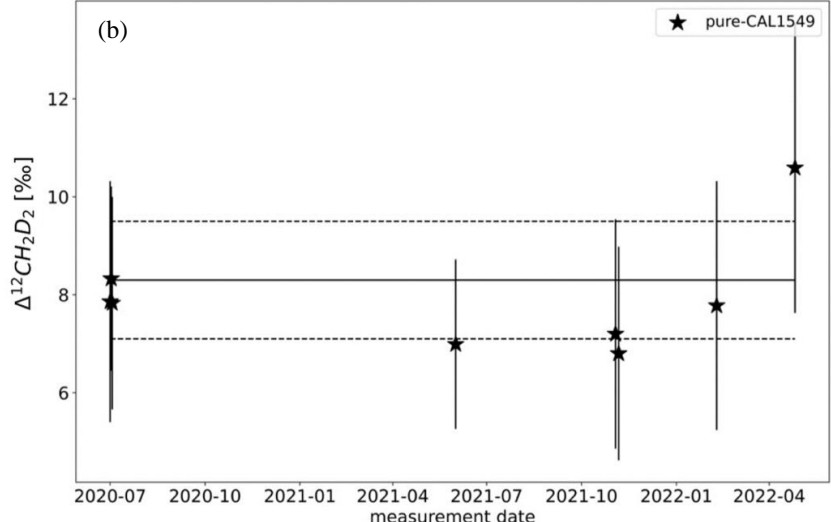

*Fig 6: Results of the measurements of pure-CAL1549 for $\Delta^{13}CDH_3$ (a) and $\Delta^{12}CD_2H_2$ (b).*
       *The solid black line represents the average value of these measurements, and the black*
       *dashed line is the standard deviation (1 σ) of the 8 measurements shown.*



The reproducibility of the measurements on the Ultra was quantified by repeated
measurements of pure-CAL1549 as shown in Fig 6. Long-term reproducibility, estimated as 1
σ standard deviation of the measurements of pure-CAL1549 over almost 3 years, is around
0.15 ‰ for $\Delta^{13}CDH_3$ and 1.2 ‰ for $\Delta^{12}CD_2H_2$. This external reproducibility is consistent
with the individual measurement uncertainty, which is on average 0.3‰ for $\Delta^{13}CDH_3$ and
2.3‰ for $\Delta^{12}CD_2H_2$ for these measurements.


### 3.4 Inter-laboratory calibration

Three of our gases, AP613, CAL1549 and IMAU-3 were measured on both Thermo Ultra at
Utrecht University (UU) and Nu Panorama at University of Maryland (UMD). The results of
these measurements are given in Table 5.

*Table 5: Comparison of $\Delta^{13}CDH_3$ and $\Delta^{12}CD_2H_2$ measurements of the three reference gases:*
*AP613, CAL1549 and IMAU-3 on the Ultra at UU and the Panorama at UMD.*


| Gas | $\Delta^{13}CDH_3$ UU (‰) | sd | $\Delta^{13}CDH_3$ UMD (‰) | sd | $\Delta^{12}CD_2H_2$ UU (‰) | sd | $\Delta^{12}CD_2H_2$ UMD (‰) | sd | $\Delta^{13}CDH_3$ difference (‰) | $\Delta^{12}CD_2H_2$ difference (‰) |
|---|---|---|---|---|---|---|---|---|---|---|
| AP613 | 2.23 | 0.12 | 1.9 | 0.5 | 3.12 | 0.9 | 3.1 | 0.8 | 0.3 | 0.02 |
| CAL1549 | 6.4 | 0.4 | 6.1 | 0.5 | 8.3 | 2.0 | 10.0 | 0.8 | 0.3 | -1.7 |
| IMAU-3 | 2.5 | 0.3 | 1.8 | 0.5 | 0.4 | 1.2 | -0.7 | 0.7 | 0.6 | 1.1 |

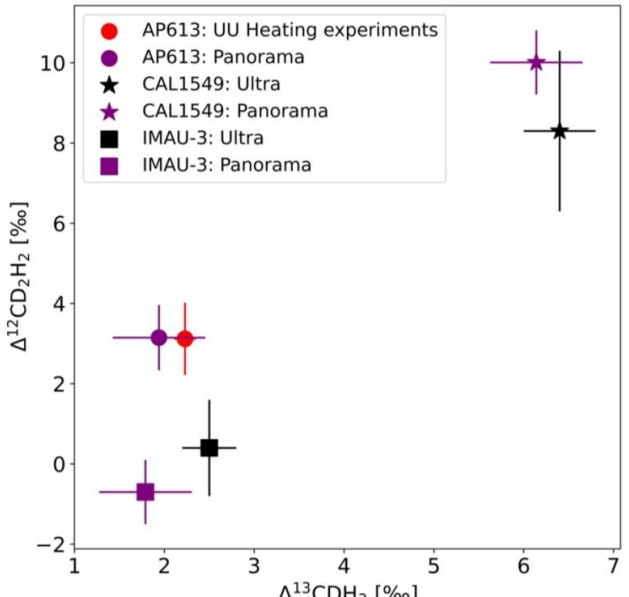



*Fig 7: The clumping anomalies of AP613, CAL-1549 and IMAU-3 measured on the Ultra (UU) and the Panorama (UMD). The red dot represents the values of AP613 obtained from our calibration experiments (as mentioned above). The black square and star represent the Ultra measurements of CAL-1549 and IMAU-3 respectively. The purple dot, star and square are Panorama measurements of AP613, CAL-1549 and IMAU-3 respectively.*


The values assigned to AP613 using our heating experiments (explained above) agree well with the values measured on the Panorama as shown in Fig 7. The other two gases are also within the measurement uncertainty (1 σ).

**3.5 Extraction test with known gas**

As mentioned earlier, mixtures of pure $CH_4$ from AP613 or CAL1549 with zero air were used to test and characterize the extraction system. The $CH_4$ extracted from these mixtures was measured against the AP613 reference gas on the Ultra. The results of the measurements are
presented in Fig 8 as the difference between the expected and the measured value. We expect this difference to be zero within the measurement uncertainty if the extraction went well and didn't cause any isotopic fractionation. Pure $CH_4$ from CAL1549 was also passed through the extraction system (hereby denoted as pure-CAL1549 extracted) using the normal extraction procedure to check for any contamination or fractionation associated with gas
introduction and collection via the extraction system.

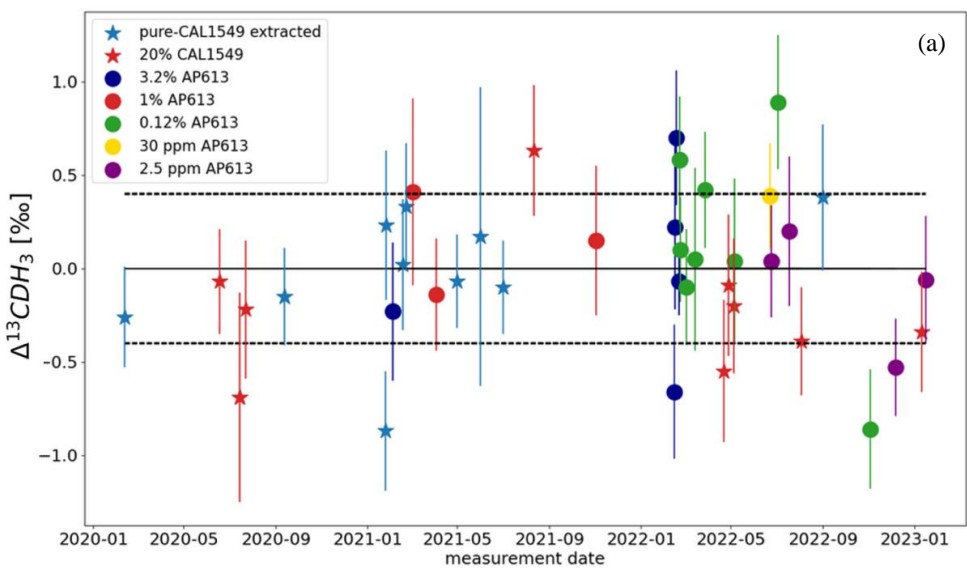



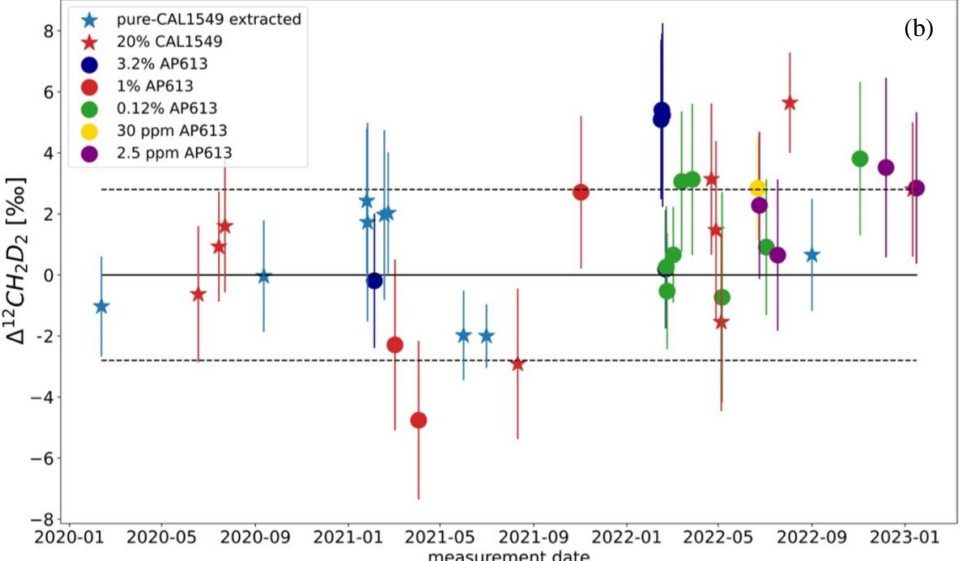


*Fig 8: Test results of the extraction system with different mixtures of laboratory reference gasses as stated in the legend. Each coloured dot and star represent the difference between the measured and expected $\Delta^{13}CDH_3$ (a) and $\Delta^{12}CD_2H_2$ (b) values, respectively, of extracted-AP613 and extracted-CAL1549 as given in the legend. The black dashed line is the standard*

*deviation (1 σ) of the difference for $\Delta^{13}CDH_3$ and $\Delta^{12}CD_2H_2$ respectively.*

The standard deviation of the difference between the expected and the measured values of these extraction tests are 0.4 ‰ for $\Delta^{13}CDH_3$ and 2.8 ‰ for $\Delta^{12}CD_2H_2$. Most of these extracted reference gas measurements are within this unexpected uncertainty (1 σ). When the

difference was more than about 2 σ, additional tests were performed, or parts of the system were replaced or cleaned longer until the measurements were good enough.

The effect of Kr on the measurements were investigated using a 1:1 mixture of IMAU-3 and pure Kr. This mixture was directly measured on the Ultra and compared with the values of

pure IMAU-3. The $\delta^{13}C$, $\delta D$, $\Delta^{13}CDH_3$ and $\Delta^{12}CD_2H_2$ of the mixture measured on the Ultra are -34.6 ‰, -242.0 ‰, 7.45 ± 0.37 ‰, 65.7 ± 2.3 ‰, respectively, whereas that of pure IMAU-3 are -36.6 ‰, -200.0 ‰, 2.5 ± 0.3 ‰, 0.4 ± 1.2 ‰, respectively. This shows that Kr introduces a strong bias on the measurements of both the bulk and clumped isotopic composition of $CH_4$. Therefore, it is very important to remove Kr from the sample before

measuring the $CH_4$ isotopic composition on the Ultra.

### 3.6. Chromatograms

Accurate and precise measurements of $\Delta^{13}CDH_3$ and $\Delta^{12}CD_2H_2$ on the Ultra requires 3 ± 1

mL of pure $CH_4$. $CH_4$ from sample mixtures pre-concentrated in the extraction system is separated from the bulk sample using the GC, as explained in detail above. Chromatograms





for samples with different $CH_4$ concentrations are illustrated in Fig 9. Depending on the $CH_4$ concentration, the total volume of the sample injected into the GC is different. When the total sample volume is above 100 mL, $O_2$ and $N_2$ are not completely separated from $CH_4$ and

therefore, a second round of GC purification is needed. For atmospheric $CH_4$ samples, separation of Kr from $CH_4$ is attained only when the GC columns are kept at 40 °C instead of the usual 50 °C used for other $CH_4$ samples.

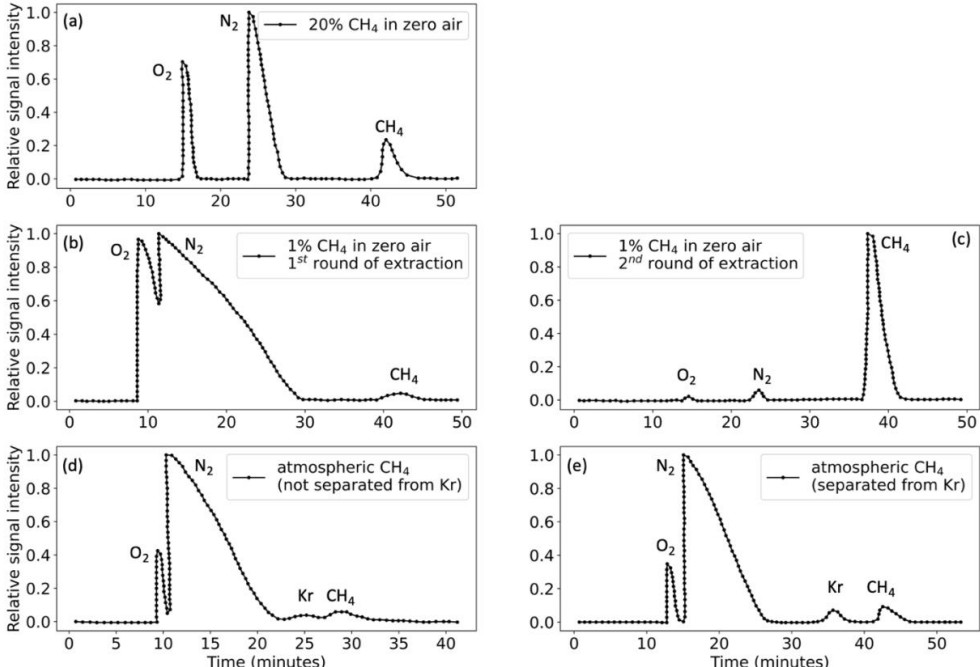


*Fig 9: GC chromatograms of different sample mixtures as shown in the legends. (a) chromatogram of 20 % $CH_4$ + 80 % zero air: 25 mL sample volume (5 mL $CH_4$). (b) and (c) chromatograms of first and second round of 1 % $CH_4$ + 99 % zero air: 250 mL sample volume (2.5 mL $CH_4$). (d) chromatogram of a pre-concentrated atmospheric air: 70 mL*

*sample volume (2 mL $CH_4$), when GC columns were heated at 50 °C and Kr is not separated from $CH_4$. (e) chromatogram of pre-concentrated atmospheric air when GC columns were heated at 40 °C and Kr and $CH_4$ are well separated.*

**3.7 Propagation of error from clumping anomaly to the formation temperature**


The clumping anomalies, $\Delta^{13}CDH_3$ and $\Delta^{12}CD_2H_2$, can be used to calculate the formation temperature of $CH_4$ when it is formed in thermodynamic equilibrium. The average precision of the Ultra measurements is 0.3 ‰ $\Delta^{13}CDH_3$ and 2.4 ‰ for $\Delta^{12}CD_2H_2$. When propagated into the calculated temperatures, the measurement error has a non-linear effect across the

temperature range of 0–1000 °C. This is because of the polynomial function that defines the relation between the clumping anomalies and temperatures as given in Equation 4a and 4b.





Figure 10 shows that the formation temperatures can be predicted with relatively low uncertainty at lower temperatures. For example, at 50 °C the formation temperature can be estimated as $50^{+13}_{-12}$ °C from $\Delta^{13}CDH_3$ and $50^{+19}_{-17}$ °C from $\Delta^{12}CD_2H_2$. At 400 °C, for the

same measurement precision, the temperature estimated from $\Delta^{13}CDH_3$ is $400^{+90}_{-66}$ °C and from $\Delta^{12}CD_2H_2$ is $400^{+410}_{-154}$ °C. Although the absolute clumped isotope effects are larger for $\Delta^{12}CD_2H_2$ than for $\Delta^{13}CDH_3$, formation temperatures calculated from $\Delta^{13}CDH_3$ give a more precise temperature estimate because of the better measurement precision for $\Delta^{13}CDH_3$.


*Fig 10: Error in the formation temperatures calculated from $\Delta^{13}CDH_3$ (a) and $\Delta^{12}CD_2H_2$ (b). The black solid line represents the thermodynamic equilibrium curve, and the blue dashed lines give the upper and lower limits of the errors of temperatures propagated from the errors in the measured clumping anomaly.*


### 3.7 Overview of different samples measured.

### 3.7.1 Samples with different source signatures

CH$_4$ samples collected from different origins and from laboratory experiments were extracted and measured with the setup explained in section 2.4. An overview of bulk and clumped isotopic composition of some of these samples from different sources of CH$_4$ is presented in Fig 11. The precision of individual measurements is 0.2 - 0.5 ‰ for $\Delta^{13}CDH_3$ and $1.4 – 4$ ‰ for $\Delta^{12}CD_2H_2$ depending on the sample volume.


Most of the samples of thermogenic origin lie on or close to the thermodynamic equilibrium line and therefore, the formation temperature of CH$_4$ can be calculated for them. All the samples with a microbial origin (e.g., incubation experiments with methanogens, CH$_4$ from lake water and sediments) have depleted $\Delta^{12}CD_2H_2$ values. The low-temperature abiotic CH$_4$

also has negative $\Delta^{12}CD_2H_2$. This is in line with previous studies that also show that





production of $CH_4$ by methanogens and in rocks abiotically at lower temperatures is affected by kinetic fractionation and/or combinatorial effect that leads to negative $\Delta^{12}CD_2H_2$. So far, we have measured about 80 samples on the Ultra from very different origins and a wide range of clumping anomalies: -1 – 6 ‰ for $\Delta^{13}CDH_3$ and -40 – 45 ‰ for $\Delta^{12}CD_2H_2$.


### 3.7.2 Ambient air measurements

Using the low-concentration extraction system (LCES), we extracted and measured several samples of atmospheric air sampled in Utrecht and the results of the first measurements are

given in Table 6.

*Table 6: Results of $\delta^{13}C$, $\delta D$, $\Delta^{13}CDH_3$ and $\Delta^{12}CD_2H_2$ of atmospheric $CH_4$ (air A, B and C) sampled in Utrecht and the comparison of the measured values to the model predictions in Haghnegahdar et al. (2017) and Chung and Arnold (2021).*


| Samples measured/ Model predictions | $\delta^{13}C$ (‰) | $\delta D$ (‰) | $\Delta^{13}CDH_3$ (‰) | se | $\Delta^{12}CD_2H_2$ (‰) | se |
|---|---|---|---|---|---|---|
| air A | -48.11 | -80.3 | 0.1 | 0.4 | 41.7 | 2.6 |
| air B | -47.99 | -84.5 | 1.87 | 0.3 | 40 | 2.5 |
| air C | -49.84 | -115.7 | 1.91 | 0.4 | 42.3 | 3.8 |
| Haghnegahdar et al. (2017) | | | 4.6 | | 114 | |
| Chung and Arnold (2021) | | | 3.3 | | 93 | |

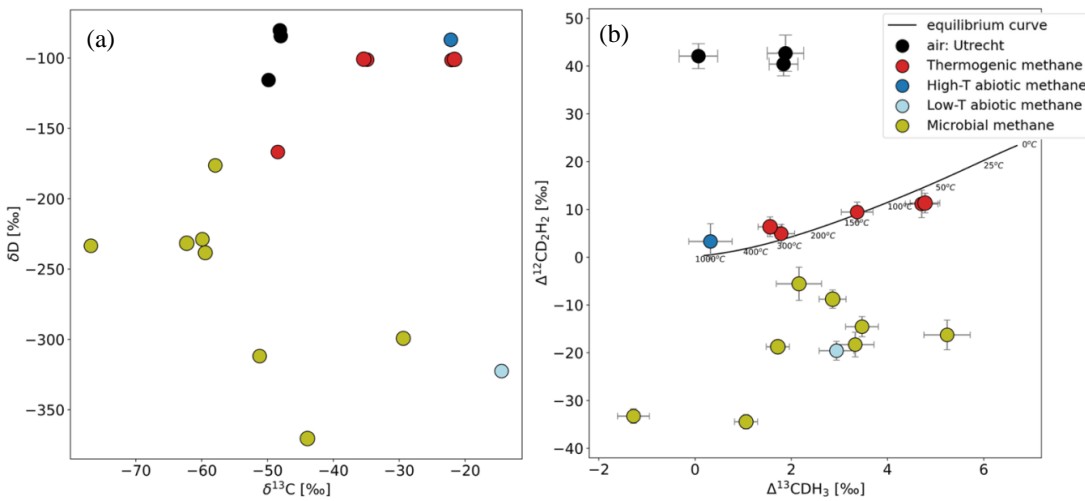



*Fig 11: Comparison of $\delta^{13}C$ and $\delta D$ (a) and $\Delta^{13}CDH_3$ and $\Delta^{12}CD_2H_2$ (b) of samples from different source types and atmospheric air measured outside IMAU. The solid black line represents the thermodynamic equilibrium curve with corresponding temperature values.*

The solid black dots in Fig 11b show the results of the first measurements of the clumping
anomaly of atmospheric $CH_4$ in Utrecht. The bulk isotopic composition of these measurements agrees well with the values reported in previous studies. Smaller air samples were collected in bags during the extraction process to be measured with the conventional continuous flow IRMS system for bulk isotopes (Menoud et al., 2020). $\delta^{13}C$ and $\delta D$ measured in both instruments agree well within the measurement uncertainty. For the
clumped isotopologues, the values of air are characterised by a very high anomaly for $\Delta^{12}CD_2H_2$ and a low anomaly for $\Delta^{13}CDH_3$. Comparing these values to $CH_4$ emitted from various sources, it is evident that atmospheric $CH_4$ has a distinct clumped signature, particularly in $\Delta^{12}CD_2H_2$. The large positive anomaly for $\Delta^{12}CD_2H_2$ of atmospheric $CH_4$ can be explained by a strong clumped isotope fractionation due to the sink reactions of $CH_4$ in the
atmosphere (Haghnegahdar et al., 2017). The distinct differences between various source types, and the offset of atmospheric $CH_4$ also suggest that more measurements of the clumping anomaly of air, especially $\Delta^{12}CD_2H_2$, can provide more information about the different sources and sink reactions that determine atmospheric $CH_4$ levels.

Although atmospheric $CH_4$ has very high $\Delta^{12}CD_2H_2$ compared to the emissions from sources, our measurement results are still far lower than recent model predictions (Chung and Arnold, 2021; Haghnegahdar et al., 2017) (Table 6). The difference can be either due to the inaccuracy in (i) source signatures of all the different sources that contribute to atmospheric $CH_4$ mole fraction (ii) the theoretical values of kinetic isotopic fractionation factor (KIE) of
the sink reactions of $CH_4$ with OH and Cl and the soil sink reactions.

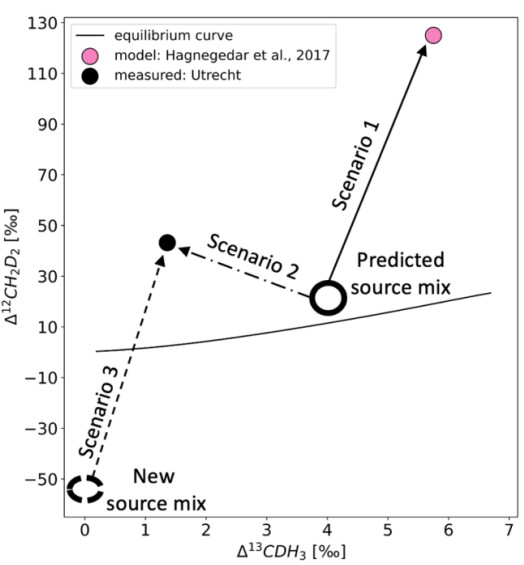



*Fig 12: $\Delta^{13}CDH_3$ versus $\Delta^{12}CD_2H_2$ space showing the different scenarios discussed. The solid*
*black line represents the thermodynamic equilibrium curve. The pink dot is the value of air*
*predicted in from the source mix shown as the solid black circle. The black dot is the value of*
*air measured on Ultra. The three arrows show the three scenarios as mentioned in the text.*
*The dashed black circle is the new source mix calculated using Scenario 3.*

We used a box model to see how the clumping anomaly of air reacts to these two parameters.
The model uses clumping anomalies of the source mixture and the KIEs of OH and Cl sinks
as input and gives the expected anomalies of air as output. We work with three scenarios as
discussed in detail below and illustrated in Fig 12.

Scenario 1: Replicating the values in the study of Haghnegahdar et al. (2017). If we assume
that the predicted clumping anomaly of the mixture of sources in the atmosphere ($\Delta^{13}CDH_3$
= 4 ‰, $\Delta^{12}CD_2H_2$ = 20 ‰) is accurate, then our model also gives higher values of $\Delta^{12}CD_2H_2$
and $\Delta^{13}CDH_3$ of air as in that study, with the same KIE used (OH: 1.92 for $^{12}CD_2H_2$, 1.33 for
$^{13}CDH_3$ and Cl: 2.2 for $^{12}CD_2H_2$, 1.46 for $^{13}CDH_3$). This was done to verify that our simple
model works well for this study.

Scenario 2: Calculating the KIEs required to arrive at the measured values of air with the
same source mix as used in Haghnegahdar et al. (2017). To get the measured values from the
predicted source mix, the KIEs must be lowered to 1.79 for $^{12}CD_2H_2$ and 1.325 for $^{13}CDH_3$
for reaction with OH and 1.9 for $^{12}CD_2H_2$ and 1.45 for $^{13}CDH_3$ for reaction with Cl. This
relatively small change causes a difference of about 60 ‰ in $\Delta^{12}CD_2H_2$ between the two
scenarios 1 and 2. Therefore, the clumping anomalies are very sensitive to the KIEs of the
sink reactions.

Scenario 3: Calculating the clumping anomaly of the source mixture that is consistent with
the KIEs used in Haghnegahdar et al. (2017) and the atmospheric air measurements presented
here. In this case, the clumped isotope anomaly of the source mixture must be heavily
depleted, especially in $\Delta^{12}CD_2H_2$ ($\Delta^{13}CDH_3$ = 0‰, $\Delta^{12}CD_2H_2$ = -54‰) to get the measured
values using the KIEs in scenario 1. This is much lower than the predicted value and would
imply a strong underestimation of $CH_4$ sources with depleted clumping anomalies such as
biogenic sources.

Given the rather high amount of clumped isotope measurements of $CH_4$ sources that have
been published to date, it seems unrealistic that the clumping anomaly of the source mix is so
depleted in $\Delta^{12}CD_2H_2$ as calculated in scenario 3, which would imply that the KIE was
previously indeed overestimated. These simple isotope mass balance calculations show that
we need very precise estimations of the sink KIEs and more accurate measurements of the
sources to completely understand the atmospheric $CH_4$ budget using clumping anomalies.




## 4. Summary and Conclusion

We have presented a new versatile analytical setup for extraction, sample preparation and measurement of the clumped isotope composition of $CH_4$ on the Thermo Ultra instrument, including samples at atmospheric concentration. The extraction and GC purification techniques do not cause significant isotopic fractionation and preserve the signatures of the $CH_4$ source. Currently, the system has been tested and works well for sample volumes of upto 1100 L. The typical precisions of samples measured on the Ultra are $0.3 \pm 0.1$ ‰ for $\Delta^{13}CDH_3$ and $2.4 \pm 0.8$ ‰ for $\Delta^{12}CD_2H_2$. The long-term reproducibility, obtained from repeated measurements of pure CAL1549 over almost 3 years, is around 0.15 ‰ for $\Delta^{13}CDH_3$ and 1.2 ‰ for $\Delta^{12}CD_2H_2$. The standard deviation of the difference between the expected and the measured values of all the extraction tests performed are 0.4‰ for $\Delta^{13}CDH_3$ and 2.8 ‰ for $\Delta^{12}CD_2H_2$. The total measurement time is around 20 hours. The system and the measurement procedure can be adjusted to optimise the sample volume required and long measurement times. First measurements of samples from various sources yield results in general agreement with published values. We have measured about 80 samples on the Ultra from very different origins and a wide range of clumping anomalies: -1 – 6 ‰ for $\Delta^{13}CDH_3$ and -40 – 45 ‰ for $\Delta^{12}CD_2H_2$. Our measurements of atmospheric $CH_4$ show enriched $\Delta^{12}CD_2H_2$ values, but not as high as recently predicted by clumped isotope models. It is unlikely that the discrepancy can be explained only by an underestimation of sources with negative $\Delta^{12}CD_2H_2$, but we show that a small adjustment in the KIEs of the sinks could reconcile atmospheric and source clumped isotope compositions. The precision of atmospheric $CH_4$ measurements can still be improved by extracting $CH_4$ from much larger samples (2000 L).

## Data availability

Data supporting this study are openly available at: Sivan, Malavika. (2023). Extraction, purification, and clumped isotope analysis of methane (Δ13CDH3 and Δ12CD2H2) from different sources and the atmosphere [Data set]. Zenodo.
https://doi.org/10.5281/zenodo.8269713

## Competing Interests

Some authors are members of the editorial board of journal AMT.

## Author contribution

All authors contributed to the design of the study. M. Sivan undertook the laboratory work with help from C. van der Veen and M. E. Popa. M. Sivan wrote the manuscript with input from all co-authors.



## Acknowledgements


We would like to acknowledge the contributions of Dipayan Paul and Sönke Szidat for their
important pioneering steps in testing the extraction line and setting up the Ultra measurement
procedure. We would like to thank Edward Young for his help building the high
concentration extraction line. We would also like to thank Jiayang Sun and James Farquhar
from the University of Maryland for the measurements of our reference gases on the Nu
Panorama instrument. We also thank our collaborators for contributing to the sample set used
in this paper. M. E. Popa, M. Sivan and the Thermo Ultra instrument are supported by the
Netherlands Earth Science System Center (NESSC), with funding from the European Union's
Horizon 2020 research and innovation programme under the Marie Skłodowska-Curie, grant
agreement No 847504.

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
