# Peer review of "Extraction, purification, and clumped isotope analysis of methane $(\Delta^{13}CDH_3 \text{ and } \Delta^{12}CD_2H_2)$ from sources and the atmosphere"

_EGUsphere, 2023_

## Author Comment (AC1)

Reply to Reviewer 2

**Thank you for your valuable feedback and concrete comments. The comments (in black) have been copy-pasted here with the answers (replies) below each (in blue). The answers specify the updated line numbers corresponding to the revised manuscript.**

This paper presents the development of 1) a measurement method for methane isotopologues including isotopologues bearing 2 heavy isotopes using a high-resolution mass spectrometer and its validation, 2) the collection and purification of CH4 from samples at a range of concentrations. The methods are then applied to perform measurements of CH4 in atmospheric air. Both the technical developments, which are very well detailed, and the natural sample measurements are valuable contributions to a growing field. I have a number of comments and suggestions, but they are altogether of limited impact on the paper, and I recommend acceptance of the manuscript with minor revisions.

Comments:

The atmospheric CH4 measurements should be mentioned in the abstract! It is one of the very first study to do so, and this should be given more visibility. This is made all the more important by the contrast with the expected clumped values from modelling, which, if confirmed as not a local effect (see next comment), will challenge assumptions about the understanding of the methane budget and motivate further investigation.

The following addition has been made to the abstract. In addition, the atmospheric samples measured are discussed in more detail in the manuscript (see below).

Line 36-37: This paper highlights the extraction and one of the first global measurements of the clumping anomalies of atmospheric methane.

The results on atmospheric CH4 deserve more discussion. The bulk isotope values measured for the three samples of Utrecht air cover a 2 permil range in d13C and a 30 permil range in dD. The range in ?13CDH3 is also larger than the analytical uncertainty (but with no correlation to the bulk values). This should be commented on or discussed, especially since the values are compared to the modelled average atmospheric values. Are those ranges meaningful or not? Are there local sources or differing wind patterns that could cause such variations in Utrecht air CH4? This can potentially inform sampling strategies.

The three air samples shown in Fig 11 were indeed sampled under three different atmospheric conditions (see below). However, all three values are much lower than the modelled average atmospheric values. In the meantime, since our paper was submitted, Haghnegahdar et al. (2023) published clumping anomalies of air (0-3 ‰ for $\Delta^{13}CDH_3$ and 42-55 ‰ for $\Delta^{12}CD_2H_2$) measured in and around Maryland, sampled under many different atmospheric conditions. Their measurement results are similar to ours, i.e. much lower than the model predictions. Therefore, local atmospheric effects being the reason for the discrepancy between our measured values and model results can be discarded.

Considering the comments from both reviewers stating the importance for further discussion of the air samples, section 3.8.2 has been modified as follows:

Line 786-814: The solid black dots in Fig 11b show the results of the first measurements of the clumping anomaly of atmospheric $CH_4$ in Utrecht (0-2 ‰ for $\Delta^{13}CDH_3$ and 40-43 ‰ for $\Delta^{12}CD_2H_2$). The air samples in Table 6 were sampled under 3 different atmospheric conditions: (i) clean air from the north (air A); (ii) clean air from the south (air B) and (iii) air with high $CH_4$ content due to local/regional pollution (air C). The values of the clumped isotopic composition of all three air samples are characterised by a very high anomaly for $\Delta^{12}CD_2H_2$ and a low anomaly for $\Delta^{13}CDH_3$. First measurements of atmospheric methane reported by Haghnegahdar et al. (2023) of air sampled from various atmospheric scenarios in and around Maryland, the USA are compatible (0-3 ‰ for $\Delta^{13}CDH_3$ and 42-55 ‰ for $\Delta^{12}CD_2H_2$) with our measured values.

Firstly, comparing these values to the ones of $CH_4$ emitted from various sources, it is evident that atmospheric $CH_4$ has a distinct clumped signature, particularly in $\Delta^{12}CD_2H_2$. The large positive anomaly for $\Delta^{12}CD_2H_2$ of atmospheric $CH_4$ can be explained by a strong clumped isotope fractionation due to the sink reactions of $CH_4$ in the atmosphere (Haghnegahdar et al., 2017). The distinct differences between various source types, and the offset of atmospheric $CH_4$ also suggest that more measurements of the clumping anomaly of air, especially $\Delta^{12}CD_2H_2$, can provide more information about the different sources and sink reactions that determine atmospheric $CH_4$ levels.

Secondly, the bulk isotopic composition (Table 6) shows as expected lower values for the polluted air C compared to the clean air A and B, indicating regional contributions from biogenic sources as is typical for the Netherlands (Röckmann et al., 2016, Menoud et al., 2021). However, in the case of the clumped isotopes, the air from the north is quite different in $\Delta^{13}CDH_3$, while the values for the polluted and clean air from the south are not very different, unlike the bulk isotopes. At this point we cannot draw strong conclusions, as we only have one measurement per condition and no information on the potential variability. More measurements of $\Delta^{13}CDH_3$ and $\Delta^{12}CD_2H_2$ of air are needed to understand if short-term local / regional atmospheric changes affect the clumping anomaly of air.

There is no mention of possible tailing from the adduct 13CH5 on the measurement position chosen for 12CD2H2, despite the very close proximity. Previous studies using an Ultra, albeit in HR, not HR+ mode, have had to include an ion correction. If the problem is absent in HR+ mode for your instrument, it would be useful to state and show it explicitly. Fig 3(d) shows the detail of the relevant peak shape, but it is not evident by eye that the contribution from the adduct tailing is negligible.

There is indeed a possible tailing from the $^{13}CH_5$ peak influencing the $^{12}CD_2H_2$ peak. The HR+ mode used for our measurements minimize the size of the tail compared to the HR mode. The reasons why we do not include an ion correction are as follows:

- We noted that for the same source pressure, the differences in the adduct peak (cps) between different gases were negligible. Our measurement is done at an approximately constant pressure, same for the sample and reference, so the size of the adduct peak is similar, and thus its influence will largely cancel out.

o   The size of the adduct peak however depends mostly on the source pressure during measurement. The influence of the adduct peak size on the $\Delta^{12}CD_2H_2$ results was tested by measuring the same gas at different source pressures (same pressure for sample and reference). We did not find any dependence of the results on the source pressure.

o   Moreover, additional tests were performed by slightly varying the measurement position within the $^{12}CD_2H_2$ peak (red dotted line in Fig 3d). The influence of the adduct is larger on the left side of the $^{12}CD_2H_2$ peak and decreases as we move rightwards. Again, no influence on the final $\Delta^{12}CD_2H_2$ results was observed.

We concluded that our results are not influenced significantly by the tailing of the $^{13}CH_5$ peak, thus no correction is needed.

Line by line minor comments/typos/etc:

l45 missing () around Li et al.

Corrected.

L56-59 no order to the references either chronological or alphabetical? This is present in other parts of the manuscript (e.g., l73-74, l78). Usually, the publication's guide style will suggest one or the other.

Corrected.

L66-67 it would be good to include a supporting citation

Added.

Line 61-62: The main $CH_4$ sink in the troposphere is photochemical oxidation by OH and Cl radicals (Khalil et al., 1993).

L63-68 presents three groups for sources, but the figure 1a that supports the following discussion of the sources contains 4 groups (which are more for accumulations or seeps of methane from the crust than for the atmosphere).

We have measured several abiotic methane samples and therefore included that category in Fig 1. Abiotic methane has also been added to the sources in line 61.

L96-97 the sentence is a bit tautological: clumping anomaly is a measure of clumping of heavy isotopes. It could be better for the non-familiar reader to replace "clumping" in the second part. I would suggest signature or value rather than anomaly in general. Deviation from the stochastic may be better (especially for methane where we see positive and negative clumping commonly).

The manuscript has been edited as follows:

Line 92-95:  The clumping anomalies, denoted as $\Delta^{13}CDH_3$ and $\Delta^{12}CD_2H_2$ are a measure of the deviation of the number of clumped molecules present relative to that expected from the random distribution of the light and heavy isotopes over all isotopologues of $CH_4$.

L112 m/z is incorrect here as you are not speaking about the ions but the molecules.

Corrected.

L123: this is very important, but the number of counts is not illustrative for someone not very familiar with the issue, especially if not compared to the count rate on the major beam. Consider giving the abundance ratios of the isotopologues instead.

The manuscript has been changed as follows:

Line 119-124: The natural abundance of the clumped molecules is very low i.e., about $4.9*10^{-6}$ and $7.8*10^{-8}$ of the total $CH_4$, for $^{13}CH_3D$ and $^{12}CH_2D_2$, respectively. The corresponding ion currents are proportionally low, typically around 6000 cps for $^{13}CH_3D^+$ and 100 cps for $^{12}CH_2D_2^+$. The cumulated number of counts control the limits of the achievable precision for the rare isotopologues. Therefore, to achieve permil-level precision, the isotopologue ratios need to be measured for a long time.

l125 and in the rest of the manuscript: mL is given for the required size of the sample. If it is mL STP, this should be written explicitly (at least the first time in the manuscript).

Corrected as follows:

Line 124-125: This requires several mL (1mL (STP) = ~45 μmol) of pure $CH_4$ for one measurement.

Line 369-371: Throughout this paper the quantity of gas is specified in mL (at STP unless otherwise specified; the conversion to molar units is: 1 mL = ~45 μmol).

l137: the sources for figure 1b are not referenced here or in the caption and encompass more than the ones at l136.

The following changes have been made to the entire paragraph for better clarity:

Line 134-147: A number of studies have reported the $\Delta^{13}CDH_3$ and $\Delta^{12}CD_2H_2$ of $CH_4$ from various sources, e.g. natural gas seeps, rice paddies and wetlands, lake sediments, shale gas, coal mines, natural gas leakage, laboratory incubation experiments (Wang et al., 2015; Young et al., 2017; Stolper et al., 2018; Loyd et al., 2016; Ono et al., 2021; Giunta et al., 2019). A general overview of the expected clumped isotope signatures of $CH_4$ from different sources is illustrated in Fig 1b. Thermogenic $CH_4$ is usually formed in thermodynamic equilibrium and therefore lies on the thermodynamic equilibrium curve between 100-300 °C. Biogenic $CH_4$ production, denoted as methanogenesis in Fig 1b, is often characterised by dis-equilibrium $\Delta^{12}CD_2H_2$ values due to the kinetic isotopic fractionation associated with methanogenesis

and/or combinatorial effects (Röckmann et al., 2016). The reported range of values for abiotic (produced at high and low temperatures) and pyrogenic $CH_4$ is also shown in Fig 1b. The predicted clumping anomaly of the atmospheric $CH_4$ source mix resulting from the combination of all sources is about 4 ‰ for $\Delta^{13}CDH_3$ and 20 ‰ for $\Delta^{12}CD_2H_2$, as reported by Haghnegahdar et al. (2017) (Fig 1b).

l141: Yeung 2016 GCA paper could be referenced as well here

Reference added.

l142 repetition of l136-137c

This line has been omitted from the manuscript.

l153 missing "a" between require and fewer

Corrected.

l181-183: the long-form explanation of "stochastic" would be more useful in the introduction where it was used on its own.

The word stochastic has been omitted and Line 92-95 has been rephrased as stated above.

l208: the aperture trims the beam in the Y-dimension while the slits do that in the X-dimension. The sentence may lead the reader to think that the aperture is located close to the other slits which is not correct.

Edited in the manuscript as follows:

Line 209-212: An additional 'aperture' option can be turned on to achieve even higher resolution (HR+), wherein the focused ion beam is trimmed further in the Y axis by an additional slit situated just before the electromagnet.

l223: resolution and mass resolving power are used interchangeably. It could be useful to give a reminder of how the mass resolving power is calculated.

Line 228: mass resolving power (5-95%).

Table 1: measurement times for each configuration would be useful in this table too.

A new column has been added to Table 1 with the measurement time.

l251: what motivates measuring alternatively and not each in one block? flexibility for differences in sample sizes, countering potential drift, or some other reason?

Because of the potential drift of the collectors, the position of the peaks needs to be periodically checked and adjusted during the measurement. This is done using the peak centre correction feature of Qtegra using the $^{13}CDH_3^+$ peak. The correction cannot be done separately for the $^{12}CD_2H_2+$ peak as it is a tiny flat peak not entirely separated from $^{13}CH_5$

adduct (Fig 3). When measuring alternately, we can get the peak centre correction parameter during the $^{13}CDH_3$ measurement, and then apply it also to the following $^{12}CD_2H_2$ measurement block.

The advantages mentioned by the reviewer, i.e. flexibility for differences in sample sizes and countering potential drift, are also valid although not the main reason.

l278: I do not understand the sentence. The adjustment is checked on the signal intensity, not the pressure, so should the tolerance not refer to the signal?

We use a modified script in Qtegra to carry out a 'continuous pressure adjustment' for the bellows. The bellow pressure corresponding to a specific signal intensity is assigned in the LabBook. During the measurement, the software checks the bellow pressure before each data point is measured and when the pressure is outside the tolerance limit (0.5 mbar), the bellow is compressed by 0.5%. This cycle is continued 5 times until the set pressure value is reached. This ensures that the gases are measured at the same pressure throughout the measurement.

l315: the procedure described is for the ?-Al2O3 in Eldridge, they used Ni on silica-alumina rather than Pt for the high temperature equilibration. Do you mean that the Pt catalyst was also activated with the same procedure?

Yes, we used the same procedure as described in Elridge et al., 2019 to activate both the catalysts, $\gamma$-$Al_2O_3$ and Pt on $Al_2O_3$.

l366: it's the measurement or the determination that takes 3mL, not the composition(s)

It is the volume of gas required for the precise measurements of the clumped isotopic composition. The manuscript has been changed as follows:

Line 368-369: Precise measurements of the clumped isotopic composition of $CH_4$ on the Ultra requires about $3 \pm 1$ mL of pure $CH_4$ for a single measurement.

Fig2: mention in the caption that acronyms are in the main text.

Added to the caption.

l383: avoid LN2 or define it first. It is a technical rather than scientific term (which would be N2, l).

LN2 has been replaced by liquid $N_2$ throughout the manuscript.

l389 should be Å rather than A°

Corrected.

l395-396: apparent contradiction for the elution times of N2, I'm guessing 22 and 35 are the start and end of its elution?

Yes, that is correct. The word 'complete' added to that sentence to clarify this.

Line 401-402: After the complete elution of $N_2$ (35 min), Trap B with silica gel is cooled with liquid $N_2$ to collect $CH_4$ for about 15 min.

l409: strictly speaking most samples will have Kr, but it is only for atmospheric samples that its amount is of the same magnitude as CH4's.

Agreed. Since the concentration of Kr is approximately half as much as $CH_4$ in atmospheric samples, the separation was not attained at 50 °C.

l410: any reason not to use 40C for all runs? the elution time for CH4 seems to be just 5 minutes later from figure 9.

All the separations can be indeed done at 40 °C. However, it was operated at 50 °C for all the samples before atmospheric samples were tested. The specific example shown in Fig 9e is only 5 minutes longer than the others, however, retention times can be much longer depending on the conditioning of the GC columns i.e., even 10-20 minutes longer if the columns were conditioned for longer, typically done when samples with higher $CO_2$ and/or impurities are extracted.

l436: would the system be able to detect a CH4 breakthrough or are there reasons to think this is unlikely to occur?

In LCES, the exhaust of the low vacuum pump is connected to the Picarro instrument. The $CO_2$, $CH_4$, and $H_2O$ mole fractions are monitored continuously throughout the extraction procedure to ensure no breakthrough of these gases from the traps. The system was tested and the capacity of CT1 was determined as 1100L of air.

l450: "is" should be "has been"

Corrected.

l451: PS4 should be P4

Corrected.

l476: for the average reader it would be useful to define or reference what is meant by zero air.

The composition of zero air added.

Line 483: zero air (synthetic air, $O_2+N_2$)

l501: ""1s standard error" do you mean that when you write 1s it is one standard error (rather than one standard deviation?). If so, it should read 1s (standard error). Same later in the manuscript.

Changed throughout the manuscript.

l515 doted should be dotted

corrected.

l556: AP613 is described as a source of gas here, rather than as a gas before in the manuscript. Is it because here you mean that it was used as the source to produce the equilibrated aliquots? This seems the case from l568, please rephrase.

Yes. The following correction has been done.

Line 544-547: The equilibrated gas (subsample of AP613 heated at different temperatures (section 2.3)) was measured against the non-equilibrated gas from AP613 (directly from the cyclinder), which is the Ultra reference gas.

l590-591: I think most readers are going to be confused about why higher count rates lead to more deviation from the shot noise limit, it would be better to develop here. Eldridge et al 2019 used data filtering because the noise was not gaussian, did you need to use anything similar? if not, do you have hypotheses other than not being perfectly on the flat top for the extra noise?

No, we do not use any mathematical data filtering method to remove the noise. However, all the data points are manually checked (not through the inbuilt Qtegra software) and the outliers (> 2 standard deviation) are removed.

The plausible reasons for the larger deviation from the shot noise limit for the $^{12}CH_3D$ measurements are as follows (also added to the manuscript). We note that measuring $^{12}CH_3D$ with the H4-CDD is not optimal from the following point of view but is still better than with the H4 – Faraday cup, and it did not work well with the wider detectors due to poor peak separations.

Line 580-589: The high-count rates (order of $10^5$) of $^{12}CH_3D$ measured using the H4-CDD detector, are close to the upper limit of the CDD operating range, and not in the optimal region.  Therefore, we expect here a lower signal-to-noise ratio (= a higher relative error). The peak top of $^{12}CH_3D$, which is not very flat and sometimes rounded, suggest that the ion beam is slightly too wide for H4-CDD with a very narrow collector list, which is not unexpected given the relatively high abundance. That means, very slight variations in the ion beam direction can result in relatively large variations in the quantity of ions entering the detector. However, the changes in $\delta D$ between different samples are much higher than the achieved precision, which is better than the one for conventional CF-IRMS instruments.

l592-593: but the calculation of the ? values requires a really precise dD, which is the main motivation for the long measurement of 12DH3/12CH4 here compared to conventional.

Yes, that's true. The precision achieved now (~2 times worse than what is expected from counting statistics) is good enough and doesn't contribute significantly to the error of the calculated clumping anomalies.

l620-624: the structure is a bit confusing at the first read, please rephrase. Ultimately it is also just a statement that your 0-enrichment gives you the expected result on average.

The text has been reformulated as follows:

Line 615-623: The results of the zero enrichment measurements using AP613 are shown in Fig 5. The mean of these measurements done over 3 years is 2.3 ± 0.1 ‰ for $\Delta^{13}CDH_3$ and 3.2 ± 0.3 ‰ for $\Delta^{12}CD_2H_2$ and all the data points fall symmetrically around the values of AP613 calibrated based on the heating experiments (2.2 ± 0.1 ‰ and 3.1 ± 0.9 ‰ for $\Delta^{13}CDH_3$ and $\Delta^{12}CD_2H_2$ respectively). The standard deviation of these measurements, 0.4 ‰ for $\Delta^{13}CDH_3$ and 2.1 ‰ for $\Delta^{12}CD_2H_2$, is close to the typical measurement error. Together, these measurements show that there are no other large sources of errors in the sample measurements (e.g., leaks in the inlet and/or room temperature variations) and that both bellows used for the measurements behave similarly.

Fig7: the legend is a bit confusing; it would be better to remove "heating experiment" here I think

The figure (and legend) has been changed.

[Figure]

*Fig 7: The clumping anomalies of AP613, CAL-1549 and IMAU-3 measured on the UU-Ultra (black) and the UMD-Panorama (purple). The shapes dot, star and square represent the gases AP613, CAL-1549 and IMAU-3 respectively.*

l695-696: it would be useful to know which parts of the system were likely to cause large offsets

The following sentence has been added:

Line 688-691: Typically, large offsets from the expected values are caused by incomplete trapping and releasing of gas from the silica gel used in Traps A and B of HCES. This is solved by conditioning the silica gel for longer (than the standard procedure, section 2.4.1) at 150 °C.

l715: some repetition of Section 2's text around this line.

The line has been omitted.

l742: it is the (temperature-induced) signal to noise ratio that is critical here rather than just the measurement precision.

Yes, that is true as stated in the previous line that the polynomial relation between the temperature and clumping anomalies is the reason for higher uncertainties at higher temperatures. In this section, we just wanted to show the limits of temperatures calculated using the clumping anomalies, both $\Delta^{13}CDH_3$ and $\Delta^{12}CD_2H_2$, associated with our measurement precision.

l769: depending on the font used it can be hard to differentiate - and —, consider using from X to Y instead of —?

The sentence has been rephrased as follows:

Line 763-765: So far, about 80 samples have been measured on the Ultra from very different origins with clumping anomalies ranging from -1 to 6 ‰ for $\Delta^{13}CDH_3$ and -40 to 45 ‰ for $\Delta^{12}CD_2H_2$.

Fig 11 I have not found a complete table of the sample compositions plotted in fig11. It would be good to at least have them in supplementary material.

All the data used for the figures were already presented in the data set (link in Data Availability). An additional table has now been added to the supplementary material and cited in Fig 11 and associated text.

References:

Giunta, T., Young, E. D., Warr, O., Kohl, I., Ash, J. L., Martini, A., Mundle, S. O. C., Rumble, D., Pérez-Rodríguez, I., Wasley, M., LaRowe, D. E., Gilbert, A., and Sherwood Lollar, B.: Methane sources and sinks in continental sedimentary systems: New insights from paired clumped isotopologues 13CH3D and 12CH2D2, Geochimica et Cosmochimica Acta, 245, 327-351, https://doi.org/10.1016/j.gca.2018.10.030, 2019.

Haghnegahdar, M. A., Schauble, E. A., and Young, E. D.: A model for 12CH2D2 and 13CH3D as complementary tracers for the budget of atmospheric CH4, Global Biogeochemical Cycles, 31, 1387-1407, https://doi.org/10.1002/2017GB005655, 2017.
Haghnegahdar, M. A., Sun, J., Hultquist, N., Hamovit, N. D., Kitchen, N., Eiler, J., Ono, S., Yarwood, S. A., Kaufman, A. J., Dickerson, R. R., Bouyon, A., Magen, C., and Farquhar, J.: Tracing sources of atmospheric methane using clumped isotopes, Proceedings of the National Academy of Sciences, 120, e2305574120, doi:10.1073/pnas.2305574120, 2023.

Khalil, M. A. K., Shearer, M. J., and Rasmussen, R. A.: Methane Sinks Distribution, Atmospheric Methane: Sources, Sinks, and Role in Global Change, Berlin, Heidelberg, 1993//, 168-179,

Loyd, S. J., Sample, J., Tripati, R. E., Defliese, W. F., Brooks, K., Hovland, M., Torres, M., Marlow, J., Hancock, L. G., Martin, R., Lyons, T., and Tripati, A. E.: Methane seep carbonates yield clumped isotope signatures out of equilibrium with formation temperatures, Nature Communications, 7, 12274, http://doi.org/10.1038/ncomms12274, 2016.

Ono, S., Rhim, J. H., Gruen, D. S., Taubner, H., Kölling, M., and Wegener, G.: Clumped isotopologue fractionation by microbial cultures performing the anaerobic oxidation of methane, Geochimica et Cosmochimica Acta, 293, 70-85, https://doi.org/10.1016/j.gca.2020.10.015, 2021.

Röckmann, T., Popa, M. E., Krol, M. C., and Hofmann, M. E. G.: Statistical clumped isotope signatures, Scientific Reports, 6, 31947, http://doi.org/10.1038/srep31947, 2016.
Stolper, D. A., Lawson, M., Formolo, M. J., Davis, C. L., Douglas, P. M. J., and Eiler, J. M.: The utility of methane clumped isotopes to constrain the origins of methane in natural gas accumulations, Geological Society, London, Special Publications, 468, 23-52, http://doi.org/10.1144/SP468.3, 2018.

Wang, D. T., Gruen, D. S., Lollar, B. S., Hinrichs, K. U., Stewart, L. C., Holden, J. F., Hristov, A. N., Pohlman, J. W., Morrill, P. L., Könneke, M., Delwiche, K. B., Reeves, E. P., Sutcliffe, C. N., Ritter, D. J., Seewald, J. S., McIntosh, J. C., Hemond, H. F., Kubo, M. D., Cardace, D., Hoehler, T. M., and Ono, S.: Methane cycling. Nonequilibrium clumped isotope signals in microbial methane, Science, 348, 428-431, http://doi.org/10.1126/science.aaa4326, 2015.

Young, E. D., Kohl, I. E., Sherwood Lollar, B., Etiope, G., Rumble, D., Li, S., Haghnegahdar;, M. A., Schauble;, E. A., McCain;, K. A., Foustoukos;, D. I., Sutclife;, C., Warr;, O., Ballentine;, C. J., Onstott;, T. C., Hosgormez;, H., Neubeck;, A., Marques;, J. M., Pérez-Rodríguez;, I., Rowe;, A. R., LaRowe;, D. E., Magnabosco;, C., Yeung;, L. Y., Ash;, J. L., and Bryndzia, L. T.: The relative abundances of resolved $^{12}CH_2D_2$ and $^{13}CH_3D$ and mechanisms controlling isotopic bond ordering in abiotic and biotic methane gases, Geochimica et Cosmochimica Acta, 203, 235-264, https://doi.org/10.1016/j.gca.2016.12.041, 2017.

---

## Author Comment (AC2)

<h1 style="text-align:center; color:blue;">Reply to Reviewer 1</h1>

**Thank you for your valuable feedback and concrete comments. The comments (in black) have been copy-pasted here with our answers in blue. The answers specify the updated line numbers as in the revised version of the manuscript.**

The manuscript reports methodological contributions in three parts:

1. collection and purification of $CH_4$ out of air, down to ambient concentrations, out of volumes up to 1100 L.
2. Measurement of $\delta^{13}C$, $\delta D$, $\Delta^{13}CH_3D$, and $\Delta^{12}CH_2D_2$ by high resolution isotope ratio mass spectrometry on the Thermo MAT253 Ultra
3. Validation, standardization, long-term repeatability, and calibration of a temperature scale with internal measurements.

Then, it contributes to the experimental literature with the measurement of clumped isotopologues in $CH_4$ in atmospheric air, and a simple box model interpretation of these results and their implications.

In all these categories, the manuscript represents a very valuable contribution to the literature. In particular, the thoroughness of the methodological description will be valuable to many future investigators in the field. And the measurement of atmospheric samples reported here, represents an early contribution to what will be an important global dataset, measured by various techniques and labs, of the atmospheric $CH_4$ clumped anomaly, putting source and sink estimates to use to constrain budgets of this important greenhouse gas. In light of its thoroughness and significance, this manuscript should be **accepted**, with some **minor revisions** as outlined below.

**Concrete comments:**

(1) The measurement of atmospheric samples that forms the culmination of this manuscript should be highlighted more specifically in the abstract and the introduction.

The following addition has been made to the abstract and introduction:

Line 36-37: This paper highlights the extraction and one of the first global measurements of the clumping anomalies of atmospheric methane.

Line 156-157: This paper presents one of the first measurements of the clumping anomalies of atmospheric methane and provide a detail comparison to the previously reported model predictions.

(2) The authors should ensure that the methods part of the manuscript is written in a way suitable for a scientific publication, as opposed to an internal protocol. The level of detail is very welcome, but care should be taken, for instance, that terms such as silica gel are written consistently (and not as Si-gel) and common abbreviations like LN2 are defined at their first mention.

"Si-gel" has been changed to "silica gel" and "LN2" to "liquid N2" all throughout the manuscript.

(3) Not all readers are necessarily conversant in the expression of gas amounts as mL at STP, so the assumptions of this nomenclature should be introduced, and perhaps a conversion to molar units given.

The following corrections were made to the introduction and methods to include the molar values:

Line 124-125: This requires several mL (1mL (STP) = ~45 μmol) of pure $CH_4$ for one measurement.

Line 369-371: Throughout this paper the quantity of gas is specified in mL (at STP unless otherwise specified; the conversion to molar units is: 1 mL = ~45μmol).

**Line-by-line comments**

97: the use of '-' in this sentence, where it could also be mistaken for a negative sign, is confusing, as is the use of both 'and' and '&' to divide parts of the sentence. Consider rephrasing.

That sentence has been omitted from the manuscript and the following addition has been made:

Line 92-95: The clumping anomalies, denoted as $\Delta^{13}CDH_3$ and $\Delta^{12}CD_2H_2$, are a measure of the deviation of the number of clumped molecules present relative to that expected from the random distribution of the light and heavy isotopes over all isotopologues of $CH_4$.

123: The ion currents of the rare isotopologues are indeed the relevant parameter for what precision is achievable, but so early in the manuscript would benefit from some more explanation of what controls these numbers.

Edited in the manuscript (also considering the Reviewer 2's suggestion) as:

Line 119-124: The natural abundance of the clumped molecules is very low i.e., about $4.9*10^{-6}$ and $7.8*10^{-8}$ of the total $CH_4$, for $^{13}CH_3D$ and $^{12}CH_2D_2$, respectively. The corresponding ion currents are proportionally low, typically around 6000 cps for $^{13}CH_3D^+$ and 100 cps for $^{12}CH_2D_2^+$. The cumulated number of counts control the limits of the achievable precision for the rare isotopologues. Therefore, to achieve permil-level precision, the isotopologue ratios need to be measured for a long time.

208: it is not clear to me from this sentence where this additional aperture is, and how it compares to the exit slit at detector H4 mentioned in line 225

Edited in the manuscript as follows:

Line 209-212: An additional 'aperture' option can be turned on to achieve even higher resolution (HR+), wherein the focused ion beam is trimmed further in the Y axis by an additional slit situated just before the electromagnet.

228: avoid use of 'etc' as far as possible—if there are any other important contaminants, it is better to be specific.

Adjusted throughout the manuscript.

248: Incorporating these measurement times in Table 1 could be useful to the reader

An additional column has been added to Table 1 with the measurement times.

350: In Fig. 2, it is not clear to me what darker and lighter connecting segments signify.

Thanks for noticing this. The apparent difference in the connecting segments was likely caused because of the image formatting. The figure has been re-edited to have the same width for all connecting segments.

406: the phrase 'larger sample size of the bulk air' is ambiguous to me

The sentence has been reformulated as follows:

Line 408-410: For samples with $CH_4$ concentrations between 1 % and 5 % $CH_4$ in air, the sample volumes required to extract the required amount of $CH_4$ are larger (>100 mL). In this case the $O_2$ and $N_2$ peaks are not fully resolved, and not well separated from $CH_4$.

410: what are the other consequences of running GC columns at 40°C instead of 50°C? Why could this temperature not be used for all samples?

All the separations can be indeed done at 40 °C. However, the GC was operated at 50 °C for all the samples before the atmospheric samples were tested. The specific example shown in Fig 9e is only 5 minutes longer than the others, however, retention times can be much longer depending on the conditioning of the GC columns i.e., even 10-20 minutes longer if the columns were conditioned for longer, which is typically done when samples with higher $CO_2$ and/or impurities are extracted.

438: are there no issues with ice clogging the first glass trap when water is frozen out of air at -70°C?

The glass trap is long, and the water freezes only at the bottom. The schematic for GT in Fig 2 has been changed to clarify this and depict the real scenario.

440: has the need for two RDTs been quantified, or is this just a factor of safety?

This system was originally designed for CO measurements, which is very sensitive to even minute amounts of remaining $CO_2$. The first RDT removes about 99.9 % $CO_2$, therefore, to ensure 100% removal, a second RDT was added to the initial system (Brenninkmeijer, 1993).

451: is PS4 the gauge labeled 'P4' in the drawing?

Corrected.

515: "dotted"

Corrected.

556: clarify that the heated gas is a subsample of AP613, which is at present ambiguous

Corrected as follows:

Line 544-547: The equilibrated gas (subsample of AP613 heated at different temperatures (section 2.3)) was measured against the non-equilibrated gas from AP613 (directly from the cyclinder), which is the Ultra reference gas.

591: why would high count rates on a CDD lead to divergence from the expected error estimate?

The following addition has been made to the manuscript to explain this:

Line 579-587: Typically, $\delta D$ measurements are about 2 times worse than the shot noise limit. This may be because of the following reasons: The high-count rates (order of $10^5$) of $^{12}CH_3D$ measured using the H4-CDD detector, are close to the upper limit of the CDD operating range, and not in the optimal region. Therefore, we expect here a lower signal-to-noise ratio (= a higher relative error). Additionally, the peak top of $^{12}CH_3D$, which is not very flat and sometimes rounded, suggest that the ion beam is slightly too wide for H4-CDD with a very narrow collector slit, which is not unexpected given the relatively high abundance. That means, very slight variations in the ion beam direction can result in relatively large variations in the quantity of ions entering the detector.

655: in Fig 7 it is ambiguous whether AP613 or AP613 that has been equilibrated at UU was measured on the UMD Panorama.

The non-equilibrated AP613 was measured on the UMD Panorama. Fig 7 has been changed as follows:

[Figure]

*Fig 7: The clumping anomalies of AP613, CAL-1549 and IMAU-3 measured on the UU-Ultra (black) and the UMD-Panorama (purple). The shapes dot, star and square represent the gases AP613, CAL-1549 and IMAU-3 respectively.*

References

BRENNINKMEIJER, C. A. M. 1993. Measurement of the abundance of $^{14}$CO in the atmosphere and the $^{13}$C/$^{12}$C and $^{18}$O/$^{16}$O ratio of atmospheric CO, with application in New-Zealand and Antarctica. *J. Geophys. Res.,* 98**,** 10,595-10,614.